# RGD-Binding Integrins Revisited: How Recently Discovered Functions and Novel Synthetic Ligands (Re-)Shape an Ever-Evolving Field

**DOI:** 10.3390/cancers13071711

**Published:** 2021-04-04

**Authors:** Beatrice S. Ludwig, Horst Kessler, Susanne Kossatz, Ute Reuning

**Affiliations:** 1Department of Nuclear Medicine, University Hospital Klinikum Rechts der Isar and Central Institute for Translational Cancer Research (TranslaTUM), Technical University Munich, 81675 Munich, Germany; Steffi.ludwig@tum.de; 2Department of Chemistry, Institute for Advanced Study, Technical University Munich, 85748 Garching, Germany; kessler@tum.de; 3Clinical Research Unit, Department of Obstetrics and Gynecology, University Hospital Klinikum Rechts der Isar, Technical University Munich, 81675 Munich, Germany

**Keywords:** RGD-binding integrins, angiogenesis, vascular normalization therapy, exosomes, synthetic integrin ligands, integrin targeted therapy, αvβ3, αvβ6, molecular imaging, SARS-CoV-2

## Abstract

**Simple Summary:**

Integrins, a superfamily of cell adhesion receptors, were extensively investigated as therapeutic targets over the last decades, motivated by their multiple functions, e.g., in cancer (progression, metastasis, angiogenesis), sepsis, fibrosis, and viral infections. Although integrin-targeting clinical trials, especially in cancer, did not meet the high expectations yet, integrins remain highly interesting therapeutic targets. In this article, we analyze the state-of-the-art knowledge on the roles of a subfamily of integrins, which require binding of the tripeptide motif Arg-Gly-Asp (RGD) for cell adhesion and signal transduction, in cancer, in tumor-associated exosomes, in fibrosis and SARS-CoV-2 infection. Furthermore, we outline the latest achievements in the design and development of synthetic ligands, which are highly selective and affine to single integrin subtypes, i.e., αvβ3, αvβ5, α5β1, αvβ6, αvβ8, and αvβ1. Lastly, we present the substantial progress in the field of nuclear and optical molecular imaging of integrins, including first-in-human and clinical studies.

**Abstract:**

Integrins have been extensively investigated as therapeutic targets over the last decades, which has been inspired by their multiple functions in cancer progression, metastasis, and angiogenesis as well as a continuously expanding number of other diseases, e.g., sepsis, fibrosis, and viral infections, possibly also Severe Acute Respiratory Syndrome Coronavirus (SARS-CoV-2). Although integrin-targeted (cancer) therapy trials did not meet the high expectations yet, integrins are still valid and promising targets due to their elevated expression and surface accessibility on diseased cells. Thus, for the future successful clinical translation of integrin-targeted compounds, revisited and innovative treatment strategies have to be explored based on accumulated knowledge of integrin biology. For this, refined approaches are demanded aiming at alternative and improved preclinical models, optimized selectivity and pharmacological properties of integrin ligands, as well as more sophisticated treatment protocols considering dose fine-tuning of compounds. Moreover, integrin ligands exert high accuracy in disease monitoring as diagnostic molecular imaging tools, enabling patient selection for individualized integrin-targeted therapy. The present review comprehensively analyzes the state-of-the-art knowledge on the roles of RGD-binding integrin subtypes in cancer and non-cancerous diseases and outlines the latest achievements in the design and development of synthetic ligands and their application in biomedical, translational, and molecular imaging approaches. Indeed, substantial progress has already been made, including advanced ligand designs, numerous elaborated pre-clinical and first-in-human studies, while the discovery of novel applications for integrin ligands remains to be explored.

## 1. Introduction

Cell adhesion to the surrounding extracellular matrix (ECM) and/or other cells constitutes the basis for the development, maintenance, and function of various cell and tissue types in multicellular organisms [1,2,3]. Since their discovery as cell-surface adhesion receptors 30 years ago, the integrin superfamily of cell adhesion and signaling receptors evolved as a key player in the link between the extracellular environment and the cytoskeleton [4,5,6]. Essential physiological processes such as embryogenesis, angiogenesis, homeostasis, hemostasis, and wound healing are mediated by integrins [1,5,7,8,9,10] as well as pathophysiological processes, such as tumor growth, metastasis, invasion, and tumor vascularization/angiogenesis. These facts and the occurrence of distinct integrin subtypes on tumor cell surfaces at elevated levels raised great hopes to classify them as valuable target structures in cancer treatment and diagnostic imaging [1,7,9,10,11].

Integrins are transmembrane heterodimers composed of one α- and one β-subunit constituting a protein family of 24 different members formed by the non-covalent combination of 18 α- and 8 β-subunits [5]. The elucidation of the crystal structure and the characterization of the extracellular domain of some family members in the beginning of the new century stimulated the generation of an enormous number of therapeutics targeting distinct integrins [12,13,14,15,16,17,18,19,20,21,22]. Docking experiments and crystal structure analyses allowed a tailor-made design of subtype-selective and highly affine integrin ligands, which is mainly based on the highly conserved recognition motif in ECM proteins, the tripeptide Arg-Gly-Asp (RGD) [4,12,13,14,15,16,17,18,19,20,21,22,23,24,25,26].

The knowledge of the biological roles of integrins and the discovery of new key functions of distinct integrin subtypes in pathophysiological processes, specifically in cancer, is continuously expanding and enables the development of new and improved integrin-targeting therapeutic strategies. To this end, significant improvements have been made in the field of synthetic integrin ligands that selectively address integrin subtypes with high affinity and specificity, which act as inhibitors and/or vehicles for drug delivery but also by agonistic properties of integrin ligands, which came recently in the focus of medicinal application [27,28,29].

In the present review, we summarize the latest discoveries in RGD-binding integrin (patho-) biology as well as progresses made in the design and application schemes of RGD-based integrin ligands for new and promising approaches in integrin-targeted therapy concepts.

## 2. Integrin/ECM Cell Adhesion and Signaling Receptors

Before we get into the role of integrins in cancer and the ligands to target them, we want to clearly lay out the basis for integrin-targeted therapy by describing the mechanisms of integrin ligand binding and the targetable vulnerabilities associated with it. Integrins are classified into four integrin subfamilies according to their binding specificity toward distinct ECM proteins [2,30]. These subfamilies are (i) the collagen-binding [5,31,32], (ii) the laminin-binding [5,31], (iii) the RGD-binding integrins [5,10,19], and (iv) the leukocyte-specific integrins that bind to intercellular adhesion molecules (ICAMs), polysaccharides, and plasma proteins, respectively [33,34,35] (Figure 1).

Most of the eight RGD binding integrins are expressed at elevated levels in various cancer types where they contribute to crucial pathophysiological functions in cancer as well as in other diseases, such as neurological disorders [36], sepsis [37], fibrosis [38], cardiovascular diseases [39], and viral infections [40]. Since this review focuses exclusively on RGD binding integrins, with respect to other integrin family members, we refer to excellent other reviews [32,41,42]. Various ECM proteins, e.g., fibronectin [43], vitronectin [44], von Willebrand factor [45], fibrinogen [46], osteopontin [47], or thrombospondin [48] encompass the RGD-motif, which is recognized by the integrin subtypes αvβ3, αvβ5, αvβ6, αvβ1, αvβ8, α5β1, αIIbβ3, and α8β1, respectively [31,34,49].

Each subunit consists of three domains: a long extracellular domain (700–1100 amino acids), a short transmembrane domain (20–30 amino acids), and a cytoplasmic region (20–75 amino acids) [2,48,50,51]. The integrin ligand-binding domain is constituted by the extracellular head domains of both integrin subunits [2,7,48,50,52]. One crucial structural element for integrin ligand binding is located within the head group of the *β*-subunit of all integrins, the so-called Metal Ion-Dependent Adhesion Site (MIDAS) [53,54,55,56]. In case of ligand binding, MIDAS is occupied by a divalent metal ion (Mn^2+^, Mg^2+^, Ca^2+^) to which the carboxy group of the integrin ligand coordinates [53,54,55,56,57,58,59]. Hence, the acidic residue represents a distinctive element of all integrin ligands [53,54,55,56,57,58,59].

In addition to mediating firm cellular adhesion to the underlying ECM, integrins bear the capacity to transmit bidirectional signals across cell membranes (Figure 2) [60,61,62,63]. This enables internal signals to control cell adhesion to the ECM (*inside-out* signaling) as well as signals from the cellular microenvironment to regulate cellular processes (*outside-in* signaling), such as wound healing, cell differentiation, migration, and proliferation [64]. This requires conformational changes in the two integrin subunits [60,61,62,63]. In the “switchblade” model [65,66,67], three integrin conformational states have been proposed, bent, extended, and extended with an open headpiece [68,69,70,71]. Already in the bent conformation, integrins are capable of binding ECM ligands with low affinity [72]. In this resting state, the transmembrane and cytoplasmic domains of the α- and the β-subunit harbor a closed conformation. Here, the helices interact in a manner, similar to that observed for the strongly homodimerizing erythrocyte protein glycophorin A (GpA), which harbors the dimerization motif GXXXG in its transmembrane domains [24,73,74,75,76]. In fact, by sequence alignments, indeed, a GXXXG-like motif was shown to be highly conserved among most integrin subunits [73,77]. Then, priming and ligand binding to integrins instigate large-scale conformational rearrangements in which the integrin extracellular domains erect [65,66,67]. During *inside-out* activation, an intracellular force stimulates cytoplasmic proteins, such as talin or kindlin, to attach to the cytoplasmic domain of the β-subunit and to destabilize a salt bridge connecting the α- and the β-subunit (Figure 2) [20,71,78,79,80,81]. The resulting separation of the transmembrane and cytoplasmic domains to an open conformation leads to a high affinity ligand-binding headpiece and now exerts integrin signaling competence [64,66,82,83,84].

The knowledge of integrin transmembrane domain conformations is indispensable for elucidating the mechanisms involved. In a cell model harboring an engineered αvβ3 transmembrane domain variant, in which the complete integrin transmembrane domains were exchanged by that of GpA in the context of an otherwise unaltered αvβ3, it was shown that this resulted in a non-signaling receptor with low affinity due to firmly associated transmembrane domains [76]. This leads to an integrin conformation with pure antagonistic function, since for signal transduction, the unclasping of the transmembrane domains is required [85]. In contrast, when the GXXXG-motif was mutated to GXXXI, which is known to abrogate dimerization, an αvβ3 variant was generated that displayed dissociated transmembrane domains, exhibiting high-affinity and constitutively active signaling competence [76]. This allows the re-association of the separated subunits to form homooligomers (dimers in the α-subunit and trimers of the β-subunits) with new RGD binding pockets [85,86], which can further bind to other proteins to form nascent adhesions (Figure 2) [20,62,69,70,71,81,87,88,89]. Subsequently, upon multiprotein clustering, focal adhesions are formed, which are the sites where integrin signaling takes place [62,68,69,70,71,88,90]. This triggers a cascade of phosphorylation events of various downstream signaling molecules, among those, most importantly the focal adhesion kinase (FAK), the mitogen-activated protein kinase (MAPK), and the extracellular signal regulated kinase (ERK) [71,75,76,81,85,91]. Obviously, multivalent binding of clustered integrins and other proteins in focal adhesions results in enhanced integrin receptor avidity and thus strong cell/ECM contacts [92]. In order to interfere with this strong cell attachment to the ECM, high amounts of ligands are required for their antagonistic action. Thus, the behavior in the activation of integrins may explain why low concentrations of the αvβ3-targeting ligand Cilengitide cause agonism, whereas antagonism needs high doses of ligands to disrupt the strong multivalent binding events within focal adhesions (see Section 4.2).

## 3. The Role of RGD Binding Integrins in Cancer and Tumor Neovascularization

### 3.1. RGD Binding Integrin Receptors

**Integrin αvβ3.** One of the earliest studied and meanwhile best characterized integrins is αvβ3. The integrin αvβ3, originally named vitronectin receptor, binds besides vitronectin also to other ECM proteins, such as fibronectin, osteopontin, von Willebrand factor, or laminin. Other integrins, such as α5β1, bind selectively to one ECM protein, e.g., fibronectin [10,93]. During tumor angiogenesis, αvβ3 is highly upregulated on angiogenic endothelial cells but not on the quiescent endothelium and mediates the interaction of activated endothelial cells with the ECM [10,93,94,95,96,97]. On tumor cells, αvβ3 is highly overexpressed in the course of tumor progression and triggers the formation of lamelli- and filopodia, regulating cancer cell migration and invasion, cell/ECM stiffness, and cell contractility [10]. Additionally, through the interaction with various ECM proteins, αvβ3 associates with Vascular Endothelial Growth Factor (VEGF) receptors and Fibroblast Growth Factor-2 (FGF-2), promoting cancer cell invasiveness [93,98]. Especially in breast cancer, αvβ3 is detected at the invasive front as well as in distant metastases [99]. In addition to breast cancer, αvβ3 is upregulated in gastric cancer, mainly in stroma and endothelial cells [10,100]. Furthermore, αvβ3 was found to be highly expressed in glioma [101], lung cancer brain metastases [102], prostate cancer [103,104], pancreatic cancer [105], oral squamous cell carcinoma [106], ovarian [10], and non-small cell lung cancer correlating with tumor progression [107,108,109]. As it is overexpressed on tumor cells, as well as the vasculature, αvβ3 can be used for the targeted therapy of both [93,110].

**Integrin αvβ5.** Integrin αvβ5 has been found to be highly upregulated in gastric tumor and stromal cells as well as in the endothelial cells of newly formed vessels during angiogenesis [100]. Furthermore, αvβ5 is overexpressed in lung cancer [102], non-small cell lung cancer [108,111], and prostate cancer [104]. In contrast to αvβ3, it promotes tumor invasion and metastasis formation rather than impacting on primary tumor growth [97,112,113]. Similar to αvβ3, αvβ5 promotes distinct pathways of angiogenesis [97]. While αvβ3 is necessary for basic FGF- or Tumor Necrosis Factor-α (TNF-α)-mediated angiogenesis, angiogenesis mediated via VEGF or Transforming Growth Factor-α (TGF-α) requires the presence of αvβ5 [97,114]. Furthermore, downstream signaling of αvβ5 leads to FAK activation as well as of Raf, leading to endothelial cell protection against apoptosis induced by ECM detachment, cellular stress, and/or DNA damaging agents [97,115]. For tumor invasion and metastasis promoted by αvβ5, in contrast to αvβ3, cytokine or growth factor stimulation is necessary [97].

**Integrin α5β1.** Another important angiogenesis marker on endothelial cells is α5β1, where it correlates with tumor malignancy, invasiveness, and thus metastasis formation [116,117,118]. During invasion processes, α5β1 facilitates the increased generation of adhesion forces, focal adhesion assembly, stress fiber formation, and contractile forces, respectively [118]. Thereby, the cell overcomes easier the steric hindrance of the ECM, leading to a higher invasiveness [118,119]. Integrin α5β1 is also expressed on tumor cells, such as oral squamous cell carcinoma [106,120] and ovarian cancer cells [117]. Furthermore, α5β1 was capable of triggering pro-survival signaling on epidermoid carcinoma cells, thereby regulating cell proliferation through both a constant activation of the Akt kinase, an apoptosis suppressor, and the Epidermal Growth Factor Receptor (EGFR) [120].

**Integrin αvβ6.** Integrin αvβ6 is exclusively expressed in epithelial cells and overexpressed in various cancer entities [10], such as oral squamous cell carcinoma [121], breast [122], colon [123], liver [124], ovarian [125], and pancreatic cancer [126,127] at the invasive front. Here, it is involved in the activation of Transforming Growth Factor-β (TGF-β1). Integrin αvβ6 activates TGF-β1 through binding to the RGD motif contained within the Latency Associated Peptide (LAP) linked to inactive TGF-β1 [128]. This mechanism works in a protease-independent fashion implicating tractional forces conferred by the mechanotransducing properties of this integrin exerted by the actin cytoskeleton. As long as LAP is non-covalently linked to inactive TGF-β1, the binding of the latter to its receptors is abrogated and prevents signaling [129]. In order to enable receptor binding, TGF-β1 has to be activated and released from this complex. Upon force application to the LAP–TGF-β1 complex, the so-called Large Latent Complex (LLC), conformational structural alterations have to occur. As a consequence, αvβ6 is a driver in epithelial mesenchymal transition (EMT) in cancer and correlates with poor patient survival as well as chemo- and radioresistance [128,130,131].

**Integrin αvβ8.** More recently, another member of the αv-integrin family, αvβ8, has been detected in head and neck cancer, non-small cell lung cancer, as well as prostate cancer [104,108,132]. It has received increasing attention in current research, since it also interacts with the RGD sequence of LAP, thereby promoting the activation of TGF-β. However, in contrast to the above described mechanism of αvβ6-mediated TGF-β1 activation, αvβ8 activates TGF-β1 in a protease-dependent way, since it harbors a shorter cytoplasmic tail, which is unable to bind to actin. Therefore, αvβ8-dependent activation of latent TGF-β1 from the LLC has been proposed to require proteolytic cleavage of LAP by co-expressed MMP-14 (or MT1-MMP) rather than by force such as in the case of αvβ6. Subsequently, TGF-β1 is released from the LLC, allowing its binding to TGF-β receptors [132,133,134]. Via its capacity to activate TGF-β1, αvβ8 is also implicated in angiogenesis, which is not the case for αvβ3 and αvβ5 [133].

**Integrin α8β1.** Integrin α8β1 is mainly expressed in smooth muscle cells [135,136] and plays a crucial role in kidney morphogenesis [137]. Its role in cancer is so far unknown. However, in murine mammary tumors, α8 was found to be co-expressed with its ligand tenascin-W, which is implicated in cell proliferation and regenerative processes [138]. Thus, in the presence of tumor stromal tenascin-W, a facilitated invasion of α8β1-expressing cancer cells into the surrounding tissue might be conceivable [138]. So far, no selective, synthetic ligands targeting α8β1 have been reported.

**Integrin αIIbβ3.** Integrin αIIbβ3 occurs exclusively on thrombocytes and serves as a fibrinogen receptor to bridge platelets together. As a consequence, αIIbβ3 represents a valuable target structure in anti-thrombosis therapy [139]. Even it does not play an active role in cancer, for the design of integrin ligands for anti-cancer therapy, it is of utmost importance to assure that these ligands do no cross-react with αIIbβ3 in order to avoid severe hemorrhagic complications in patients receiving systemically administered ligands [140].

**Integrin αvβ1.** Integrin αvβ1 has received limited attention as a disease-relevant target in the past, which was possibly a consequence of the absence of specific and selective inhibitory peptides or antibodies. It is highly expressed on activated fibroblasts and directly binds to the latency-associated peptide of TGFβ1 and mediates TGFβ1 activation [141,142]. Thus, αvβ1 has been suggested as a promising target in fibrosis treatment, but its exact role in TGFβ1 activation, compared to other integrins, remains to be dissected. Integrin αvβ1 has also been described a receptor for viral entry, for example as an adenovirus co-receptor.

### 3.2. Integrin Functions in Angiogenesis and Tumor Vessel Phenotype

#### 3.2.1. The Role of Integrins in Angiogenesis

One important cancer hallmark, in which integrins are majorly involved during tumor growth and progression, is the induction of tumor neovascularization [143,144,145]. Distinct integrin subtypes are majorly implicated in these angiogenic processes [146]. Integrin αvβ3 was the first member of this superfamily found to be overexpressed on angiogenic endothelial cells in concert with numerous pro- and anti-angiogenic factors, most importantly VEGF and its receptors [147]. Since VEGF positively affects the conformational activation state of αvβ3 and its signaling capabilities, a feed-forward loop of VEGF release by tumor cells and increased cell adhesion and migration is constituted. In turn, αvβ3 activation hampers VEGFR2 degradation, boosting continuous signal transduction even more [148]. Thus, αvβ3 was soon recognized as an important marker for angiogenesis.

In pre-clinical studies, the use of αvβ3 antagonists, such as antibodies, RGD-based cyclic peptides, or peptidomimetics proved to efficiently suppress tumor angiogenesis and disease progression. This strongly motivated their exploration in clinical cancer trials as new anti-cancer strategies [93,149]. In this regard, the first RGD-based peptide was Cilengitide (EMD 121974) [62,150], which is directed to the binding epitope for ECM protein ligands constituted by the headgroups of the two subunits of αvβ3 and αvβ5, respectively. The clinical trial experience with Cilengitide is further discussed in Section 4.2 and is the subject of a number of excellent and detailed reviews [95,151,152,153,154]. In brief, although Cilengitide exhibited a promising performance in in vitro cell studies and in pre-clinical animal models, it missed the high expectations of its therapeutic benefit in human therapy trials for patients afflicted with different cancer types [62,155,156]. In the course of the exploration of its clinical efficacy, it became obvious that treatment with low-dose Cilengitide led to unexpected pro-angiogenic effects. This observation inspired a series of interesting studies to unravel the underlying mechanisms. In fact, it was found that low concentrations of Cilengitide promoted VEGF-mediated angiogenesis, which is an effect whose mechanisms motivated further investigations [157]. In mouse endothelial cells treated with low-dose Cilengitide, an increased VEGFR2 protein expression was noticed, with no impact on VEGFR2 mRNA, indicating a post-transcriptional regulation. The same was observed in tumor blood vessels in vivo. Moreover, the exposure of cells to low-dose Cilengitide affected the cellular trafficking of VEGFR2 and αvβ3, since it was documented that the recycling of both receptors back to the cell membrane was strongly promoted, without impacting on receptor internalization. As a consequence of this enhanced recycling, a diminished VEGFR2 degradation was noticed that was concomitant with αvβ3 recruitment to focal adhesions [148,157]. The Cilengitide-triggered increase of VEGFR2 cell surface expression was traced to the level of downstream mediators leading to VEGFR2 phosphorylation as well as integrin and Src activation [157,158]. Even more, low-dose Cilengitide induced αvβ3 affinity and signaling, leading to augmented tumor cell proliferation and enhanced angiogenesis via increased VEGFR-triggered signaling and enforced endothelial cell migration [157]. In addition, Cilengitide has a short plasma half-life of 3–5 h, limiting its time for accumulation in the tumor. Hence, even though high doses were systemically applied, tumors were potentially exposed to the low-dose, pro-angiogenic effects of Cilengitide, explaining its inability to efficiently treat tumors [159]. Taken together, these Cilengitide effects are thought to explain its failure as an angiogenesis inhibitor [160]. In addition to the pro-angiogenic effects of low-dose Cilengitide in cancer, it was debated if it may also exert vascular-promoting effects in heart failure models [29]. Indeed, the treatment of mice who underwent abdominal aortic constriction surgery with low-dose Cilengitide stimulated coronary angiogenesis and affected cardiomyocyte hypertrophy, resulting in a mitigated course of the heart disease. Analyses at the molecular level revealed that low-dose Cilengitide affected the transcriptome of cardiac endothelial cells, involving pro-angiogenic signaling pathways as well as genes implicated in cell growth and DNA repair, also enhancing angiogenesis.

#### 3.2.2. The Therapeutic Benefit of Vascular Normalization Therapy in Cancer

Tumor vessels are markedly different from their normal quiescent counterparts since they display rather immature features, such as chaotic branching, collapses, and microaneurysm. Moreover, tumor vessels show incomplete endothelial cell lining, a defective basement membrane, and a lack of other cellular vessel components, such as pericytes and smooth muscle cells, which stabilize and cover normal vessel walls [94,161,162,163,164,165,166,167]. Due to those facts, no regular blood flow is established in tumor vessels, reinforcing a hypoxic environment within the tumors, fibrosis, and extravasation of tumor cells into the blood or lymphatic system, resulting in the enhanced permeability and retention (EPR) effect. This leads to the entry of macromolecules into the tumor interstitial space, missing drug delivery and therapy response, as well as infiltration by immune regulatory cells [168]. Thus, new approaches are urgently needed to address these challenges by developing a vascular promotion therapy aiming at vascular normalization. Vessel maturation is achieved by the upregulation of adhesion molecules on endothelial cells as well as by the recruitment of vascular smooth muscle cells and pericytes that interact with activated endothelial cell and contribute to vessel branching and coverage. This leads to increased tumor perfusion, oxygen, and pH levels, the recruitment of immune effector cells, as well as improved drug delivery and thus therapy response (Figure 3). Thus, tumor-associated angiogenesis is the net result of a complex and fine-tuned interplay between cancer cells, stimulated vascular and perivascular cells, recruited myeloid and mesenchymal stem cells, as well as activated fibroblasts and immune cells within the tumor microenvironment [169,170]. These events involve integrin crosstalk with the VEGFR2 and the Platelet-Derived Growth Factor Receptor (PDGFR) [171,172].

Once it was discovered that low-dose treatment with Cilengitide exerted pro-angiogenic effects in tumors, an alternative use of low-dose Cilengitide was inspired in order to increase blood flow and, consequently, increased delivery and intracellular uptake of chemotherapeutic drugs [157]. These observations changed the paradigm in cancer therapy from anti-angiogenic strategies to vascular promotion/normalization therapy [173,174]. In mouse and human cancer models, the administration of Cilengitide in combination with Verapamil, a calcium channel blocker, which increased vessel dilation and blood flow in tumors by relaxing blood vessel muscles, enhanced gemcitabine delivery to the tumor bed and thus the efficacy of this drug to provoke tumor regression and metastasis suppression [27]. Moreover, vascular promotion therapy allowed the administration of significantly reduced doses of gemcitabine, thereby lowering side effects and enabling longer treatment windows [173,175,176,177,178].

#### 3.2.3. Integrins in Tumor Cell Extravasation

Leaky tumor vessels do not only exert a negative impact on drug delivery within the tumor site but also favor the escape of shed or detached tumor cells into the bloodstream and lymphatic system as a crucial step for tumor cell dissemination and metastasis. This extravasation occurs as a function of the permeability and missing integrity of the tumor vascular endothelium. Vessel wall permeability is under the control of inflammatory mediators as well as growth factors and proteases, which are sequestered by tumor cells and other cells within the tumor microenvironment. Integrins, which are expressed on endothelial cells, take over important functions since they mediate their adhesion to the basement membrane, which represents an important prerequisite for tumor vessel integrity [179]. Integrin α5 affects tumor cell extravasation by interacting with neuropilin 2 (NRP-2). NRPs are multifunctional transmembrane non-kinase receptors that are expressed on cancer cells [180]. They function predominantly as co-receptors and thus rely on other cell surface receptors to transduce transmembrane signals. The dysregulation of NRPs has been implicated in many pathological conditions, including tumor cell survival, migration, and invasion involving them in the regulation of angiogenesis [181,182,183,184]. Thus, the overexpression of NRPs provokes hypervascularization due to leaky vessel formation [185,186]. Integrins also affect angiogenesis and tumor cell extravasation by their association with NRPs, which are found in integrin-containing multiprotein complexes of adhesomes and are capable of modulating the PI3K/Akt/PTEN signaling axis by activating integrins. Hereby, *inside-out* signaling of β1, β3, and β5 integrins, respectively, is activated, leading to enhanced binding to ECM proteins [187,188,189,190]. In addition, α5β1, α6β1, and α9β1 expressed on tumor cells are capable of binding to endothelial cell-expressed NRP-2, thereby facilitating cancer cell binding to the endothelium, followed by promoted tumor cell extravasation in distinct cancer types [187,191,192,193,194,195,196]. These findings underline the importance of NRP in concert with integrins and the VEGFR for the proper development of blood vessels and their stabilization. Thus, NRPs also represent potential new therapeutic targets due to their multifaceted roles and the fact that they are highly expressed on tumor cells and tumor endothelial cells [180].

### 3.3. The Cellular Fate of Tumor Cell-Expressed Integrins

#### 3.3.1. Cellular Internalization of Integrins

Integrins undergo constant endo-/exocytosis processes, which are crucial for directed cell migration and the dynamic formation and release of adhesive cell–ECM contacts in order to move a cell’s body forward [197,198]. Thus, the balance between endocytosis and recycling of cell-surface integrin receptors is decisive for proper cell migration. It is well known that integrins are constitutively internalized through clathrin- or caveolin-mediated pathways, implicating various additional players. Alternative routes include macropinocytosis from circular dorsal ruffles upon growth factor receptor-mediated signaling [199] and endocytosis via clathrin-independent carriers. The kind of pathway that will be pursued for trafficking either via caveolae, clathrin, neither, or both, strongly depends on the respective cell type and integrin subspecies [200], the latter depending on sequence motifs contained within the highly variable C-terminal tail of β-integrins [201]. As such, in fibroblasts and fibrosarcoma cells, α5β1 internalization occurred in a clathrin- and microtubule-dependent fashion [202,203], whereas in myofibroblasts, α5β1 is constitutively taken up in a caveolin-mediated manner [204] (Figure 4).

#### 3.3.2. Intracellular Endosomal Integrin Trafficking and Sorting

Intracellular endosomal sorting decides over the fate of integrins, which may either be prone to degradation or recycling back to the cell surface [205]. Even when integrins are frequently shuttled to the degradative route initially, in many cases, they are eventually recycled back to the plasma membrane. As such, it has been found that α5β1, destined to be sorted to late endosomes or lysosomes, could still escape from this degradation route to be recycled back to the cell surface [206]. The blockade of this recycling route enhanced the degradation of internalized integrins [207,208,209,210]. In case integrin ligands are conjugated to drugs, e.g., for their α5β1-mediated intracellular delivery, a portion of the endocytosed integrin was recycled, even from lysosomes, while the drug was selectively degraded.

Integrin recycling occurs by spatially and temporally different pathways [211,212,213]. In principal, two different recycling routes have been discovered, the long-loop and the rapid (short) recycling track. The long-loop pathway has been described for the endocytosis of α5β1, α6β4, and αvβ3, respectively. It starts with integrin trafficking to the Perinuclear Recycling Compartment (PNRC), which is enriched by recycling endosomes carrying members of the Rab family of small GTPases, Rab11 and Rab25, as well as Arf6 [214,215,216,217,218,219]. Recycling of α5β1 is under the control of the PKB/Akt kinase pathway via distinct downstream target structures, such as the Arf6-specific GTPase Activating Protein (GAP) ACAP1, which is crucial for clathrin coat assembly [220,221]. However, certain differences regarding involved players in the recycling process have been documented. Long-loop recycling of α5β1 in fibroblasts did not depend on growth factors [214] and also β1 recycling—brought to terms by Rab11 and Arf6—was not reinforced in the presence of EGF or serum [215]. In contrast, the rapid (short) integrin recycling route from Rab4-positive endosomes was under the control of growth factors, such as PDGF, which directed αvβ3 to this route [214]. In addition, in cervical cancer cells, β1 and β3 integrins were recycled via the rapid (short) loop requiring EGF stimulation [222] (Figure 4).

Even different integrin subtypes exhibit similar preference toward distinct ECM ligands, the biological effects arising thereof turned out to be integrin subtype- and cell-type specific. For endocytosed α5β1 and αvβ3, differential routes for recycling and crosstalk have been reported. Even more, they were interdependent, since αvβ3 activation had a negative impact on α5β1 recycling [223]. Whereas αvβ3 recycling enforced the generation of lamellipodia and increased directional cell movement in a Rac-dependent manner, α5β1 recycling to invasive cellular protrusions accelerated random cell migration and invasion via the Rho–ROCK–cofilin pathway by altered actin cytoskeleton [148,224,225,226,227]. Thus, distinct functions of integrin subtypes that have to be temporally acquired by a (tumor) cell are tightly controlled by tipping the balance between integrin degradation and recycling [205]. The net effect of continued recycling from endosomes after internalization is an increased cell surface integrin receptor density and an altered membrane distribution, e.g., in newly formed focal adhesions at the leading migratory cell front. By binding to the ECM, directed and efficient cell motility was promoted, which was not the case upon constant integrin breakdown [211,214]. Moreover, there is also a need for molecules assisting in cell migration, which are also transported via endosomes. As such, Membrane Type 1 Matrix Metalloprotease (MT1-MMP) was trafficked to the cell invasive front, contributing to ECM remodeling for enhanced cell motility and chemotaxis [228].

Most interestingly, intracellular sorting of integrins is under the control of their activation state. In several cancer entities, both, active and inactive β1 receptors are internalized via clathrin- and dynamin-dependent routes to Rab4a-, Rab5-, or Rab21-positive early endosomes [229]. However, from there, different trafficking routes are being pursued: active β1 engaged by an ECM ligand was shuttled to late endosomes and lysosomes, either for degradation [230] or for ligand detachment, followed by the slow recycling of the ligand-free integrin back to the cell membrane via the Rab11-dependent pathway. In contrast, inactive unligated β1 was rapidly recycled back to Arf6-positive cell membrane protrusions in an actin- and Rab4-dependent fashion [229]. Ligation of α5β1 by fibronectin instigated the ubiquitination of the α5 cytoplasmic domain, ultimately resulting in the sorting of the α5β1/fibronectin complex to lysosomes by implicating the Endosomal-Sorting Complex Required for Transport (ESCRT) [231]. Still, the process of coordinated α5β1-provoked internalization of fibronectin as well as the secretion of de novo synthesized fibronectin is not fully resolved yet. Secreted fibronectin assembles into a fibrillar network via integrins, which triggers mechanical and chemical cues, which is also crucial for endothelial cell tubulogenesis [232]*,* in vivo angiogenesis [233,234], and the establishment of endothelial cell apico-basal polarity [235]. After endocytosis into early endosomes, active α5β1 engaged by protease-cleaved fibronectin fragments [204,236] was detected in post-Golgi carriers following its trafficking back to the endothelial cell basolateral surface by carrying de novo synthesized fibronectin dimers. The latter kept α5β1 in an activated state within post-Golgi carriers where talin and kindlins are missing. On one hand, these processes lead to the disposal of endocytosed and old fibronectin, and on the other, they lead to the deposition of newly synthesized fibronectin into the ECM to assure the dynamic renewal of polarized fibronectin fibrils. These findings emphasize a role of integrin recycling via the endosomal pathway for the control of ECM protein accumulation and remodeling, enabling tumor cell invasion into adjacent tissues [237].

Moreover, in the fateful decisions between the breakdown and recycling of integrins, endocytic adaptor proteins are involved. As such, sorting nexin 17 prevented lysosomal β1 degradation by interacting in early endosomes with the NPxY-motif contained within its cytoplasmic domain [208,238]. In addition, Rab proteins are critical for integrin recycling and trafficking processes in cancer cells [219,239,240]: (i) Rab21 recognizes a conserved motif found in the α-tails of all integrins and controls integrin endocytosis [241,242]; (ii) Rab25 is expressed in specific epithelial and some cancer tissues [243] and drives α5β1 recycling [244]; and (iii) Rab4 and Rab5 contained within early endosomes are instrumental for integrin sorting to other cellular compartments. Integrins, which are destined for decay, are sorted to the late endosome and to lysosomes, involving the substitution of Rab5 by Rab7 implementing the Homotypic Fusion and Protein Sorting (HOPS) complex. Then, Rab7 takes over its functions, leading to the maturation of early into late endosomes [245,246].

Even the Rab11 recycling machinery is instrumental for almost all integrins during their transfer from the cytoplasm to the cell membrane [247]; still, each integrin subtype harbors distinct trafficking characteristics [248,249]. This is dependent on Rab11 Family Interacting Proteins (Rab11-FIPs). For α6β1 trafficking and recycling, Rab11-FIP5 is required. Consequently, its deletion led to intracellular α6β1 accumulation and thus to its decreased cell surface expression on prostate cancer cells, which is followed by their downregulated α6β1-dependent migratory and metastatic properties [250,251]. In ovarian cancer cells, Rab11-FIP5 expression correlated with poor patient outcomes [252]. Other examples for integrin subtype-specific Rab11-FIP regulators are given for α5β1 and αvβ3, whose trafficking is mediated by Rab11-FIP1 [148]. These findings underline that Rab11-FIPs are decisive for specific recycling events of distinct integrin subtypes depending on the respective α-integrin subunit [253]. Most importantly, the effects of recycling-associated factors on the net cancer cell surface integrin distribution also decide over the possible use of respective integrin-targeted ligands for therapy and/or diagnosis, which requires a sufficiently high integrin density and accessibility.

The density of cell surface integrins also determines the successful and firm adhesion of tumor cells to the ECM at the metastatic site [254,255]. In this context, Rab11b-mediated endosomal recycling is important to allow tumor cell seeding at metastatic brain sites [254]. In search of differentially and temporally regulated genes during the formation of brain metastases, Rab11b was identified to be localized to the Endosomal Recycling Center (ERC). In breast cancer cells metastasizing to the brain, increased Rab11b levels were detected, permitting among others the recycling and adapted localization of β1. In vivo, a lack of Rab11b drastically decreased the development of brain metastases [254].

Several other regulators of integrin trafficking had been identified, disclosing Rab13 and Rab10 as interaction partners of the Golgi-localized Gamma Ear-Containing Arf-Binding Protein 2 (GGA2). Arf-binding proteins associate with clathrin-coated vesicles and localize to the trans-Golgi network and endosomes [256,257,258,259]. GGA2 interacts with activated β1 and affects its recycling. Both Rab13 silencing and GGA2 depletion led to intracellular β1 accumulation and thus to decreased β1 activity in focal adhesions and thus impaired cell migration [260]. In addition, several other multiprotein complexes have been discovered, such as the Retriever- [261] and the CORVET-complex, which determine specific integrin subtype trafficking routes [262]. CORVET is composed of the two subunits Vps3 and Vps8. For fusion events between early endosomes, these directly interact with each other, localize to Rab4-positive recycling vesicles, and co-localize on Rab11-positive recycling endosomes. If both are lacking, the delivery of endocytosed integrins to recycling endosomes and thus their subsequent return to the cell membrane is postponed. This leads to disturbed cell adhesion, focal adhesion formation, and cell migration (Figure 4) [262].

The proportion between distinct cell surface-expressed integrin subtypes strongly depends on the balance between their recycling and degradation. This exerts tremendous consequences for integrin signaling and the biological events arising thereof, not only under physiological but also pathophysiological settings. In fact, dysregulated integrin recycling was shown to affect tumor cell growth, invasion, and metastasis, but also tumor cell survival, e.g., by their escape of apoptosis [255]. The better understanding of intracellular integrin trafficking, integrin recycling, or degradation mechanisms will majorly inspire the development of inhibitors for novel and revisited integrin-targeted cancer therapeutic strategies.

#### 3.3.3. Exosomal Integrins in Cancer

For maintaining tissue homeostasis, cell/cell communication is of high importance. The discovery of extracellular vesicles has revolutionized the perception of these events. Exosomes are the smallest among those vesicles, ranging in size between 30 and 150 nm. They have been detected in a number of body fluids, including blood, urine, and amniotic fluid [263,264]. Their generation starts on early endosomes, followed by their maturation in multivesicular bodies (MVB) via the invagination of the endosomal membrane resulting in intraluminal vesicles (ILVs). Constitutive exocytosis into the extracellular microenvironment occurs in most cell types upon the fusion of ILVs with the cell membrane [263,264,265]. Exosomes are recognized as potent vehicles in intercellular crosstalk via the delivery of their cargo composed of proteins, lipids, metabolites, DNAs, RNAs, and microRNAs to recipient cells [266,267,268,269] (Figure 5).

In addition, during cancer progression and metastasis, enhanced exosome release by tumor cells governs crucial functions by allowing communication with other cells, including endothelial, mesothelial, and stromal cells, as well as fibroblasts and infiltrating immune cells, by transmitting pro-tumorigenic factors to distant sites in specifically tumor cell-targeted organs [270]. This cargo of tumor exosomes includes cell surface-anchored proteases, transmembrane receptors, growth and angiogenic factors, intercellular signaling messengers, as well as ECM molecules and integrins [271,272,273], which all contribute to the preparation and organization of a pre-metastatic niche [274,275,276] (Figure 5). To this end, tumor exosomes also take over distinct functions related to vascular leakiness, stromal cell education at organotropic sites, and bone-marrow-derived cell education and recruitment [266,276,277]. Tumor exosomes harboring e.g., αvβ5, α6β4, or α6β1, are capable of anchoring to distinct ECM proteins, such as fibronectin, vitronectin, laminin, or collagen [278]. Thus, in lung-tropic metastatic exosomes, α6β4 and α6β1 were found, and in liver-tropic exosomes, αvβ5 was found. Thus, strategies targeting tumor exosomal integrins may effectively block organ-specific metastasis [276].

Furthermore, integrins associated with extracellular vesicles may be utilized for tracing tumor cells within an organism, as has been documented for α6, αv, and β1, respectively [279]. These findings underline that circulating tumor exosomes harvested as liquid biopsy may be utilized to predict the tendency of certain tumor cells to metastasize to specific organs and thus predict local tumor cell seeding [280].

In addition, αvβ3 is shuttled via exosomes to recipient non-tumorigenic as well as tumorigenic cells, where it enhances their adhesive and migratory/metastatic properties. As such, the formation of cell membrane protrusions, such as filopodia, in the recipient cell is a consequence of exosomal delivery, endocytosis, and consecutive αvβ3 recycling to the recipient’s cell surface in a functionally active form [281]. Similarly, also, αvβ6 is exosomally delivered to prostate cancer cells, where it localized on the cell membrane provoking increased cell migratory activity on the αvβ6-specific substrate LAP of TGF-β1. Hereby, metastasis is promoted in a paracrine manner [282]. Thus, tumor-derived exosomes are important contributors to the horizontal transfer of distinct integrin subtypes to specific recipient cells thereby affecting their cellular phenotype and behavior.

#### 3.3.4. Exosomes as Therapeutic and/or Diagnostic Tools

The discovery of extracellular vesicles and exosomes and their important functions in the organism has inspired a series of concepts for their implementation as therapeutic and/or diagnostic vehicles. These include their application as circulating biomarkers to be harvested from body fluids as liquid biopsy as well as for the delivery of exosomally packaged drugs to tumor sites. Exosomes are uniquely suited for these purposes, since they are endowed with high delivery efficiency, excellent in vivo stability, and only minor cytotoxicity when compared to other currently used synthetic drug carriers [283]. Moreover, exosomes express transmembrane and membrane-anchored proteins that promote endocytosis, thereby enhancing the delivery of their internal cargo [284]. In fact, several exosomal vehicle variants have already been explored regarding their suitability for targeted drug delivery to cancer cells or cells related to other diseases: (i) native exosomes, (ii) native modified exosomes harboring ligands or receptors and carrying therapeutic agents as cargo, (iii) exosomes derived from genetically engineered cells, or (iv) exosome mimetics [285]. Via the encapsulation of drugs and/or the addition of specific ligands to the exosomal surface by chemical and physical procedures, a broad variety of specific and target-selective vehicles have been generated [286,287]. The ligands explored so far include monoclonal antibodies, proteins, peptides, small molecules, carbohydrates, and aptamers, which were designed to target different receptors overexpressed on the recipient cell membrane. Among those are human epidermal growth factor receptor 2 (HER2), VEGFR, e-selectine, nucleolin, transferrin, folate receptor, EGF receptor, and αvβ3 [288].

Regarding integrin targeting, several effective delivery platforms have been established. Prominent examples are doxorubicin-loaded exosomes, which have e.g., been decorated with RGD-based peptides on their surface and were shown to enable efficient and specific drug delivery into αv-expressing breast cancer models after intravenous injection [289]. In another example, doxorubicin-loaded exosomes were engineered to express the exosomal membrane protein Lamp2b fused to the αv-specific iRGD peptide and also showed highly efficient targeting of breast cancer models in vitro and in vivo [289]. Other types of exosomes have been used to target ischemic brain lesions in mouse models. These exosomes were surface modified with cyclo[RGDyK] and loaded with curcumin, and they were able to cross the blood–brain barrier to reduce inflammation and apoptosis [290]. Lastly, cyclo[RGDyK] decorated exosomes loaded with miR-210, a robust target of hypoxia-inducible factors (HIF), were also able to increase survival in mice after brain ischemia [291].

Whereas the role of distinct exosomal integrins has meanwhile been intensively explored, their function in recipient cells is not yet fully clear. In a recent study, a function of β3-integrin in the endocytosis of vesicles was described, emphasizing its role in exosomal intercellular communication critical for cancer metastasis. Based on earlier findings on the role of β3 in tumor stress resistance and metastasis in triple-negative breast cancer [292], its contribution to exosome biogenesis and endocytosis has been investigated [293]. Hereby, it was shown that clonal tumor cell growth stimulated by extracellular vesicles depended on the presence of β3 on the recipient tumor cell surface at the metastatic site, which permitted cell entry of the extracellular vesicles via β3-mediated internalization. By exploring the underlying mechanism, it was found that syndecans were involved in β3-mediated extracellular vesicle capture. Syndecans, heparan sulfate-decorated cell-surface proteoglycans, which recognize heparin-binding motifs, had been shown before to associate with β3 [294]. By applying heparin, which served as a glycosaminoglycan mimetic of heparan sulfate, extracellular vesicle uptake was blocked, indicating that heparan sulfate proteoglycans served as the bridge between exosomes and β3. The process was accompanied by the interaction of β3 with FAK leading to its activation and consequently, to clathrin-mediated and thus dynamin-driven endocytosis [293].

## 4. Design and Development of RGD-Based Integrin Ligands for Biomedical Applications and Translation

### 4.1. From Natural ECM Proteins to Synthethic RGD-Based Integrin Ligands: The Generation of Highly Selective and Affine Integrin Ligands

Historically, linear and cyclic peptidic integrin ligands had been designed, mimicking the RGD binding sequence of natural ECM proteins [295]. Derived from RGD-based cyclic peptides, antibodies, functionalized peptides, aptamers (short nucleic acid sequences), and non-peptidic RGD analogues had been developed as new classes of integrin binding compounds/antagonists [14,24,57,150,296,297,298,299]. Hereby, peptidomimetics, peptides, and small molecule ligands either imitate or block the biological effects of a natural ECM ligand [14,24,57,150,296,297,298,299,300]. Nonetheless, the overall goal in the design strategy was to generate highly selective ligands, which bind with high affinity to a respective integrin subtype (Figure 6). Given the number of integrin subtypes and their numerous and distinct roles in physiological and disease processes, a controlled therapeutic approach is usually aimed at highly selective ligands, while non-selective ligands could lead to strong side effects. For example, unwanted αIIbβ3 activity in cancer therapy can lead to severe hemorrhage [110,139]. Through cyclization (e.g., in case of cyclo[RGDfV]) as well as the incorporation of d-amino acids (e.g., Cilengitide), highly active and metabolically stable peptidic ligands were generated (Figure 6) [295,301,302,303]. In addition, the different modifications shown in Figure 6 may be combined, as e.g., for cyclo[RGDfV] or 29P, which harbor d-amino acids in their structures. In contrast, linear peptides seem to be rather metabolically instable and are often cleaved in vivo within a few minutes. Moreover, disulfide bridged peptides have lower in vivo stabilities than so-called *homodetic* cyclic peptides (cyclized head-to-tail). Furthermore, *N*-methylation of external and solvent-exposed amide bonds in cyclic peptides often resulted in a significant enhancement of selectivity among the group αvβ3–α5β1–αIIbβ3 [304,305,306,307]. On top of that, *N*-methylation confers better pharmacokinetic properties, such as enhancement of blood half-life in vivo (e.g., cyclo[FRGDLAFp(*N*Me)K(Ac)] (Ac = acetyl) (Figure 6) [300,305,308]. All synthetic integrin ligands share an essential acidic group for metal-coordination in the MIDAS region as a common feature, such as a carboxylate moiety, e.g., aspartate, as found in natural ECM ligands (Figure 6). Moreover, potent isosteric replacements, such as hydroxamic acids for αvβ3 and α5β1 [57], had been synthesized as well. Alongside the acidic group, which is necessary for binding to the β-subunit, a basic group is needed for optimal α-subunit interaction (Figure 6). For example, the direct introduction of an Arg (pKa ≈ 13.8) [309] or its isosteric replacement with less basic groups, such as 2-aminopyridine (pKa ≈ 7), 1,8-tetrahydronaphthyridine (pKa ≈ 7), or 2-aminoimidazole (pKa ≈ 8.5) [110,310,311,312]. However, it should be mentioned that e.g., Asp residues of different integrin α-subunits bind the guanidinium group of the Arg in the integrin ligand in a different way, e.g., in the αv integrins side-on but in the α5 integrins end-on (and side-on) [313]. Furthermore, through the isosteric replacement of the RGD mojety, peptidomimetic integrin ligands were generated, harboring higher metabolic stability and activity in comparison to linear, peptidic RGD-containing integrin ligands. For peptidic ligands in general, the basic and acidic moieties are arranged on a rigid peptide skeleton in order to achieve the most suitable integrin receptor interaction [314,315,316]. In addition, in case of peptidomimetic integrin ligands, a rational design, e.g., by docking studies, is applied for the generation of selective ligands. Furthermore, in case of αvβ3 ligands, an aromatic residue close to the acidic group can bind in a hydrophobic cleft, leading to an enhanced affinity of the integrin ligand [13,317]. The different affinities of many integrin ligands have been recently reviewed [318].

In general, many parameters have to be considered toward ligand translation into clinical use (e.g., affinity, selectivity, and pharmacokinetics). Whenever metabolic stability is required, cyclic peptides or peptidomimetics should be preferred.

### 4.2. Design of Synthetic RGD Binding Integrin Ligands

**Integrin ligands targeting αvβ3.** Due to the important role of αvβ3 in angiogenesis and tumor progression, many different ligands targeting this integrin have been designed over the past decades [19]. Nonetheless, only a few αvβ3 small molecule antagonists have entered clinical trials so far [10,110,319,320,321,322]. One of the major drawbacks of the most previously generated αvβ3 antagonists is their complex concentration-dependent receptor pharmacology. In fact, that has been uncovered after the first clinical trials with Cilengitide (cyclo[RGDf(*N*Me)V], ligand **1**, Figure 7), which is a cyclic pentapeptide designed for the treatment of glioblastoma, whose performance did not lead to the expected anti-angiogenic effects. Cilengitide represents a ligand for both αvβ3 and αvβ5 (IC_50_ αvβ3 = 0.61 nM, IC_50_ αvβ5 = 8.4 nM) [318]. Since this compound was extensively described in recently published reviews, we briefly describe some important facts regarding the clinical evaluation of Cilengitide in this review [26,323]. Based on early phase I and II clinical studies in glioblastoma patients, there was initially great hope for both Cilengitide high-dose monotherapy [324] and combination treatments [325,326]. In the end, large multicenter randomized phase II (CORE) and phase III (CENTRIC) clinical cancer trials using Cilengitide in combination with standard treatment for patients with newly diagnosed glioblastoma or by intensified doses during radiotherapy, respectively, failed to reach the primary endpoints of those studies [154,160]. This disappointing clinical efficacy of Cilengitide instigated the search for explanations for these failures, along with the exploration of alternative applications for its clinical use. Unexpectedly, the administration of low-dose Cilengitide in vitro and in animal models revealed that opposite to its intended purpose, it promoted tumor angiogenesis instead of inhibiting it [157].

Accumulating evidence indicates that at low concentrations, Cilengitide actually operates as an agonist, where it binds to the resting state of αvβ3, which harbors a bent conformation [66,157,327]. This induces conformational changes resulting in erected extracellular domains presenting a high-affinity conformation of the headpiece. Subsequently, upon further conformational changes, the two integrin legs are separated and integrin signaling is triggered (see Section 2). In contrast, at high concentrations, Cilengitide acts as an antagonist, which effectively inhibits integrin signaling. This dose-dependent functional switch was confirmed in vitro, showing that Cilengitide treatment with concentrations below the IC_50_ value showed agonistic signaling, while antagonistic effects occurred at concentrations above the IC_50_ [157,328]. How this applies to organisms in vivo, i.e., in which a systemic dose results in a “low” (agonistic) or a “high” (antagonistic) concentration at the target site (i.e., the primary tumor and/or metastasis), has not been systematically studied. In this respect, especially due to the rapid pharmacokinetics, i.e., the 4 h blood half-life of Cilengitide in humans [62,329], even high systemic doses likely result in low tumor doses several hours after drug application. For example, in the CENTRIC study (Merck), patients received 2 g Cilengitide intravenously twice weekly, which will be reduced to only 1 µg after 3.5 days, and to an even lower dose finally at the tumor site. Hence, the agonistic low dose regimen cannot be avoided in such a dose set-up [154,160]. In contrast, in case of Cilengitide, it is also published that low-dose agonistic action leads to vascular promotion effects and stimulation of VEGF-mediated angiogenesis (see Section 3.2.2) [27,157]. Recently, interesting applications of the vascular stabilization effect in immune therapy, cancer treatment, and sepsis have emerged (see Section 6.1) [27,37,328,330].

In addition to Cilengitide [62,331], this dose-dependent functional switch has also been observed for MK-0429 (ligand **2**, Figure 7) [319,320], which is an orally available αvβ3 integrin ligand that has been applied for the treatment of osteoporosis and melanoma [332,333]. The case of MK-0429 is curious, as it has initially been described to target only αvβ3, but later low nanomolar affinities with IC_50_ values of 1.6, 2.8, 0.1, 0.7, 0.5, and 12.2 nM for αvβ1, αvβ3, αvβ5, αvβ6, αvβ8, and α5β1 [333], respectively, were reported, marking MK-0429 as an αv pan-integrin antagonist.

In this regard, two pure αvβ3 antagonists, TDI-3761 (ligand **3**, Figure 7) and TDI-4161 (ligand **4**, Figure 7), were synthesized through modifying MK-0429 in such a manner that a π–π interaction with the Tyr122 residue from the β3-subunit was established, freezing the integrin in an inactive conformation, thereby abrogating agonistic actions of the ligand [328]. The phenomenon of pure αvβ3 antagonism was described in a former study, where a high affinity mutant (hFN10) of 10 kDa wild-type fibronectin (wtFN10, partial agonist) displayed pure αvβ3 antagonism [334]. Similar to synthetic ligand binding, wtFN10 binding triggers tertiary changes in the β3 subunit, which persist after the dissociation of the ligand, making wtFN10 agonistic. Contrary to this, hFN10 undergoes π–π interactions with the integrin ligand upon its binding, avoiding conformational changes within the β3 subunit [334]. TDI-3761 and TDI-4161 block αvβ3-mediated cell adhesion to ECM ligands without having an impact on integrin conformation as confirmed by antibody binding, electron microscopy, X-ray crystallography, and receptor priming studies [328].

Moreover, in contrast to treatment with low-dose Cilengitide, both compounds did not exert pro-angiogenic effects, such as aortic vessel sprouting, which could lead to tumor progression in vivo [328]. Thus, these pure antagonists represent a potential alternative to be extensively studied as anti-angiogenic therapy in tumors. Since TDI-4161 and TDI-3761 conveyed inhibitory effects on bone resorption, new treatment options for osteoporosis are conceivable [328]. Hence, recent developments show progress toward the development of pure antagonistic and pure agonistic ligands of αvβ3, which will hopefully lead to drugs for different medical use.

While the optimization of metabolic stability, selectivity, molecule rigidity, activity, bioavailability, and other factors was achieved [295,301,302,303,304,305,306,307,335,336], the lack of oral bioavailability of peptidic integrin ligands remained a major limitation for clinical use. With the synthesis of the αvβ3-selective cyclic hexapeptide 29P (ligand **5**, Figure 7), a highly systematic design strategy for the receipt of a biologically active as well as orally available peptide was described for the first time [308]. Different steps in the generation of the lead compound 29P were required. First, an identification of intestinal permeable *N*-methylated cyclic hexa-Ala peptides was necessary in order to obtain permeable peptidic backbone structures [308]. Then, by using a spatial screening approach [295], the RGD-motif was reintroduced into the cyclic hexa-Ala backbone to gain a bioactive compound. After optimizing this ligand via suitable flanking residues, the best ligands for RGD-recognizing integrin subtypes were selected via an enzyme-linked immunosorbent assay (ELISA)-like test system yielding peptide 29 with the following IC_50_ values: αvβ3—0.6 nM, αvβ5—145 nM, αvβ6—120 nM, αvβ8 > 1000 nM, and α5β1—21 nM, respectively [308,318]. However, the charged residues of Arg and Asp abrogated cell permeablility. In order to regain the cell permeability of this highly active cyclic hexpapeptide, the charged functional groups were protected by the pro-drug concept, i.e., by introducing lipophilic protection groups, yielding 29P [308]. After transport in the bloodstream, the protection groups are rapidly cleaved off, and the bioactivity of the RGD-peptide 29 is restored. 29P was subsequently analyzed in vivo regarding its oral bioavailability, using mice bearing Lewis lung carcinoma tumors [308]. Low concentrations of orally administered 29P had a similar effect as intravenously injected non-orally available 29 (29P without the lipophilic protection groups) or Cilengitide, and they resulted in enhanced VEGF-mediated angiogenesis by upregulation of the VEGFR2 expression [308]. However, recent approaches using low-dose Cilengitide indicated that its induction of vascular promotion could render it a promising agent for combination therapy, since increased tumor vessel stability can enhance the delivery of chemotherapeutics to the tumor site [27,330,337]. Consequently, 29P could be an ideal orally available peptidic pro-drug for vascular promotion therapy in combination with chemotherapy [308].

In addition, a tumor-penetrating disulfide bridge harboring peptide, called iRGD (cyclo[CRGDK/RGPD/EC]; ligand **6**, Figure 7) was identified, targeting αv-integrins specifically expressed on endothelial cells of tumor vessels, i.e., αvβ3 together with αvβ5, when injected intravenously [338,339]. After integrin binding, iRGD is proteolytically cleaved within the tumor, leading to the peptide sequence CRGDK/R [22,340]. While losing subsequently much of its integrin-binding capability, the truncated peptide CRGDK/R gained affinity for NRP-1, which is a membrane-bound co-receptor that binds among other ligands also the VEGF through the C-terminal exposure of the active C-end Rule (CendR) motif. The CendR motif of iRGD is hereby RGDK/R. In general, for NRP-1 binding, Gly and Asp can be replaced by any amino acid, whereby instead of Arg, Lys can be used: R/KXXK/R [22,340]. Via the interaction of the CendR motif with NRP-1, tumor tissue penetration is triggered, and iRGD has been documented to deeply enter into extravascular tumor tissue [22,339,340]. iRGD can be conjugated to drugs or imaging agents or co-administered with them, allowing the extravasation and penetration of the co-applied agents after the binding process of CendR and NRP-1 [338,340]. This is currently explored in a phase I clinical study, where metastatic prostate cancer patients first undergo monotherapy with iRGD, followed by a combination therapy with nab-paclitaxel and gemcitabine (NCT03517176). Furthermore, iRGD was used as carrier molecules for the transport of drugs (via nanocarriers) or imaging agents to selectively target tumor tissues. Through amidation, maleimide-thiol chemistry, *Michael* addition, and alkyne-azide click chemistry, respectively, nanoparticle surfaces of lipid micelles, Abraxane^®^ (albumin/paclitaxel nanocomplex) as well as iron oxide nanoworms were conjugated to functionalized iRGD molecules for the targeted delivery of drugs such as doxorubicin, paclitaxel, or vandetanib [341]. However, the studies resulted in a slightly lower tumor accumulation of the drugs after targeted delivery via iRGD in comparison to co-administration [341,342]. The limited number of the target receptors on the vasculature may hinder conjugated iRGD to promote an enhanced drug absorption, while unconjugated, co-administered iRGD may trigger a bulk transfer of drugs into the tumor [341].

**Integrin ligands targeting αvβ5.** A major challenge for the development of αvβ5-selective ligands is given by its close homology to αvβ3. A homology model of αvβ5 exhibited high similarity of β3 and β5, but the larger selectivity determining loop in β5 allows less space for ligands [343]. Hence, some ligands for αvβ3 do not fit into the αvβ5 receptor, but most αvβ5 ligands bind also αvβ3, leading to ligands such as Cilengitide or iRGD targeting both αvβ3 and αvβ5 [343]. Through a recent Structure–Activity Relationship (SAR) survey among a series of amide-containing 3-aryl-succinamic acid-based RGD mimetics, the α,α,α-trifluorotolyl residue containing ligand 7 (Figure 7) was discovered as a highly selective αvβ5 ligand, harboring an 800-fold lower affinity toward αvβ3 [344]. Docking experiments revealed that the amide-containing rigid core of ligand 7 represents the selectivity determining part of the molecule toward the other αv-integrins [344]. More precisely, the central aryl ring substituted by the CF_3_ group represents an ideal moiety for αvβ5 rather than αvβ3 binding [344].

**Integrin ligands targeting α5β1.** Regarding the selectivity of ligands discriminating between α5β1 and αvβ3, it was demonstrated that it may be directed through a methylation of the guanidinium group within the RGD motif. An integrin ligand binds to αv side-on, whereas it binds to α5 end-on [313]. Through methylation at the right position of the guanidinium group, either the side-on or the end-on binding can be suppressed, leading to a rise in selectivity toward α5β1 or αvβ3, which was demonstrated for Cilengitide [313]. The modification of the guanidinium group can be used as well for the improvement of the α5β1 selectivity [313].

A set of α5β1-targeting antagonistic peptidomimetics was synthesized, leading to the identification of the ligands **8** and **9** (Figure 7) as the most promising candidates [15,57,345]. Through rational design, it was tried to mimic the RGD motif by replacing Arg by a pyridine (ligand **8**) or a guanidinium group (ligand **9**) and the Asp of RGD by propionic acid (ligands **8** and **9**) [15,57,345]. The dimethylisopropoxy benzamido goup in both ligands undergoes π–π interactions with the (β1)-Tyr127, contributing to α5β1 selectivity. The obtained integrin antagonists exhibited high selectivity and a low-nanomolar affinity toward α5β1, representing a promising vehicle structure for imaging agents and drug targeting [15,57,345].

The peptide sequence *iso*DGR was first described in aged fibronectin. Hereby, *iso*D is formed from Asn (N) through an in situ rearrangement of the peptide sequence NGR [346,347,348]. The synthetic introduction of the *iso*DGR moiety as a replacement of RGD in cyclic pentapeptides of the sequence cyclo(X*iso*DGRX) (X = any amino acid) has been described to generate integrin ligands with high αvβ3 or α5β1 selectivity, depending on the introduced amino acids X [317]. Regarding αvβ3 selective *iso*DGR-derived compounds, the reader is directed to recent publications about the prospects of their future use in cancer therapy [346,349,350]. The *iso*DGR derived cyclic peptide cyclo[phg-*iso*DGRk] (phg = d-phenylglycine; ligand **10**, Figure 7) was designed as a highly selective α5β1 (IC_50_ = 8.7 nM) integrin ligand, which also displayed targeting of αvβ6 (IC_50_ = 19 nM) [317]. This ligand was further developed and transformed into the ^99m^Tc-radiolabeled compound ^99m^Tc-HisoDGR for its use as a Single Photon Emission Computed Tomography (SPECT) imaging probe of α5β1-positive glioma (see Section 5) [351,352]. Regarding cancer therapy, no attempts were published, but the use of cyclo[phg-isoDGRk] as a drug vehicle is conceivable.

**Integrin ligands targeting αvβ6.** The discovery of αvβ6 as a tumor biologically relevant and overexpressed biomarker in many cancer types was followed by the development of several highly selective ligands. The reported αvβ6 ligands, which did not show anti-cancer activity, could be especially useful as molecular imaging vehicles for diagnostic purposes, e.g., for PET imaging, and for integrin-targeted cancer therapy through conjugation to drugs or nanoparticles (NP).

A disulfide-bridged 16-amino acid peptide, SFITGv6 (amino acid sequence: GRCRFRGDLMQLCYPD, ligand **11**, Figure 7), containing the integrin targeting amino acid sequence FRGDLMQL with high affinity and selectivity for αvβ6 was identified within a Sunflower Trypsin Inhibitor-1 (SFTI-1)-based phage-displayed library [353,354]. The peptide demonstrated a homogenous binding to αvβ6 in head and neck squamous cell carcinoma (HNSCC) as well as in breast and lung cancer-derived metastases, which harbor elevated levels of αvβ6 [353]. In a first study in which HNSCC and non-small cell lung cancer patients received ^68^Ga-DOTA-labeled SFITGv6, a specific accumulation in tumors had been observed [353]. SFITGv6 may represent a promising tracer for imaging and possibly endoradiotherapy of αvβ6-positive cancers [353]. In further studies, diverse RGD-based SFTI-1 derivatives were generated, leading to the identification of another potent αvβ6-targeting peptide, SFLAP3 (amino acid sequence: GRCTGRGDLGRLCYPD), which is a ligand based on LAP3 (ligand **12**, Figure 7) [354]. In comparison to SFITGv6, SFLAP3 exhibited improved affinity for αvβ6 in HNSCC in vitro and in vivo (mouse and human), rendering it also a promising small molecule compound for cancer diagnosis as well as therapy [354].

Furthermore, the linear 20-amino acid peptide A20FMDV2 with the amino acid sequence NAVPNLRGDLQVLAQKVART (ligand **13**, Figure 7) was described as a potent αvβ6-inhibitor in vivo [355]. The sequence of the peptide derived from an envelope protein of the foot-and-mouth disease virus (FMDV), which mediates virus infection via binding to αvβ6 [355,356,357]. Within the peptide, the sequence RGDLQVL is responsible for αvβ6 targeting. A20FMDV2 was conjugated to ^18^F-fluorobenzoic acid for Positron Emission Tomography (PET) imaging of αvβ6-positive cancer in vivo (see Section 5). Its use has also been investigated for cancer therapy by conjugating the DNA-binding pyrrolobenzodiazepine-based payload SG3249 (tesirine) to A20FMDV2 [127]. In vitro studies showed the αvβ6-dependent cytotoxicity of this peptide–drug conjugate, which is called SG3299. In vivo, SG3299 treatment strongly reduced tumor growth and increased life span in mice with αvβ6-expressing xenografts. The nonapeptide cyclo[FRGDLAFp(*N*Me)K(Ac)] (Ac = acetate, ligand **14**, Figure 7) represents a highly selective and affine cyclic nonapeptide for targeting αvβ6 (IC_50_ αvβ6 = 0.3 nM) [313,358]. The cyclic pentapeptide SDM17 encompassing the sequence cyclo[RGD-Chg-E]-CONH_2_ (Chg = l-cyclohexylglycine, ligand **15**, Figure 7) also represents a highly affine αvβ6 targeting integrin ligand (IC_50_ αvβ6 = 1.3 nM) [359]. These peptidic ligands, together with other ligand architectures, e.g., linear peptides and knottin peptides, were developed as molecular imaging agents, which we review in detail in Section 5.

**Integrin ligands targeting αvβ8.** The octapeptide cyclo[GLRGDLp(*N*Me)K(Ac)] (ligand **16**, Figure 7, IC_50_ αvβ8 = 8 nM) represents a selective and highly affine αvβ8-targeting ligand, bearing the potential to be used as a drug delivery vehicle as well as an imaging agent [360]. This first cyclic αvβ8-binding peptide featured a strong discriminating power of at least two orders of magnitude against the other RGD-recognizing integrins [360]. The ligand has been used so far for tumor imaging, which is further described in Section 5.

**Integrin ligands targeting αvβ1.** The first and only peptidomimetic inhibitor for αvβ1, called “c8” (ligand **17**, Figure 7, IC_50_ αvβ1 = 0.089 nM), reduced TGF-β activation and consequently chemically induced pulmonary and liver fibrosis in mice, hence representing a promising αvβ1-targeting integrin ligand [141,142].

**Pan-integrin ligands.** A pan-inhibitor is defined as a compound that does not only display an inhibitory effect on an individual but rather on multiple integrin subtypes. In general, pan inhibitors are supposed to generate a better therapeutic outcome by the simultaneous inhibition of several integrins, compared to those targeting exclusively one integrin subtype, since it better addresses the frequently observed heterogeneity of integrin expression in tumors [110,361]. Furthermore, cancer cells were able to modulate their integrin expression pattern as a consequence of cancer treatment, which might not affect the efficacy of pan-selective ligands but could induce resistance to highly subtype-selective integrin ligands [362]. The small molecule αv-inhibitor CWHM12 represents such a pan-integrin ligand targeting both liver and lung fibrosis, which is driven through αv-integrins. It is assumed that pharmacological targeting of all αv-integrins may lead to better therapeutic effects in anti-fibrosis therapy [363]. CWHM12 seems to inhibit TGF-β binding, which is a major profibrogenic cytokine [363]. The compound MK-0429, which was mentioned in Section 4.2, is as well a small molecule αv pan-integrin ligand.

## 5. Integrin Targeted Molecular Imaging and Radiotherapy

The improved definition of the role of integrins as predictive biomarkers for response, prognosis, and even as therapeutic targets is an important subject of ongoing investigations. Nevertheless, the consistent overexpression of integrins in a large number of various tumor types strongly supports their roles as targets for molecular imaging. Somewhat independent from the biological functions of distinct integrins, their high expression levels in tumors can be exploited to visualize tumors and metastasis by diagnostic imaging. In addition, integrin imaging agents could be used to stratify patients for targeted therapies, to assess treatment response, and to monitor tumor growth. In this regard, nuclear imaging, including Positron Emission Tomography (PET) and Single Photon Emission Computed Tomography (SPECT), are the most established modalities for integrin imaging. Moreover, integrin imaging has also been described for a wide range of other modalities, e.g., fluorescence-guided surgery, optoacoustic imaging, ultrasound imaging, magnetic resonance imaging, Raman spectroscopy, and multimodal combinations thereof. The abundance of recent publications suggests continuous and intense research activities into pre-clinical development and clinical translation of integrin imaging probes, as we outlined below.

The first integrin-targeted molecular imaging approaches were focused on αvβ3, with the goal to develop companion diagnostics to obtain quantitative information on the angiogenic activity within tumors and to stratify patients via PET or SPECT imaging for αvβ3-targeted therapy. Therefore, αvβ3-targeting RGD-based peptides have been radiolabeled, e.g., with Fluorine-18, Gallium-68, Copper-64, or Technetium-99m, and entered pre-clinical and clinical evaluation [11,364,365,366,367]. So far, the collective evaluation of data from an abundance of studies revealed that imaging of angiogenesis is possible, but the interpretation of αvβ3 imaging data is complicated by the fact that αvβ3 is not exclusively expressed on newly formed tumor vessels, but also on tumor cells and macrophages, as well as often in normal tissues [368,369]. Furthermore, it was shown that angiogenesis does not depend on the presence of αvβ3 [368].

An important lesson from these early studies on integrin-targeting molecular imaging was that focusing on the angiogenic activity of tumors with αvβ3-imaging does not do justice to the multiple roles of αvβ3 and other integrins in tumor development, progression, and prognosis, so its clinical merit remains to be fully defined. The ongoing optimization of the pharmacokinetic properties of ligands for αvβ3 and the availability of selective ligands for other integrins (e.g., α5β1, αvβ6, and αvβ8; as described in Section 4.2 of this article), in conjunction with a better understanding of their expression patterns and biological functions, has certainly contributed to advances in molecular imaging of integrins. Furthermore, novel diagnostic and theranostic concepts of integrin-coated nanomaterials have emerged, such as integrin-targeted NPs for PET, ultrasound, and Raman spectroscopy. In this chapter, we will discuss novel developments in the field of molecular imaging of integrins, with particular attention on, but not limited to, studies published between 2015 and 2020.

### 5.1. Nuclear Imaging (PET/SPECT)

An extensive review on PET imaging probes summarizes the pre-clinical development and progress of 8 RGD-based PET tracers for αvβ3 imaging in clinical studies until 2016 [370]. Here, we focus on new developments in PET and SPECT imaging, not only those targeting αvβ3, but also other integrin subtypes, which might prove clinically even more useful. We will consider how recent studies are addressing challenges encountered in earlier studies and analyze additional evidence for the clinical relevance of RGD-based integrin imaging.

**Nuclear imaging of integrin αvβ3.** The earliest PET and SPECT imaging ligands consisted of radiolabeled peptide monomers, e.g., the cyclic pentapeptide ^18^F-Galacto-RGD, which was the first candidate to enter in-human imaging and showed the proof-of-principle of in vivo integrin imaging [367]. The drawback of this compound was the complex time-consuming radiosynthesis, which led to the exploration of simpler and faster radiolabeling strategies for integrin ligands (see [371] and [372] for reviews on this topic). In addition, it was also discovered in the past decade that the multimerization of RGD-based ligands can improve their affinity and tumor uptake compared to their respective monomers in PET and SPECT imaging approaches [373,374]. In this regard, it is important to note that higher affinity and tumor uptake of a ligand does not necessarily improve its imaging performance (e.g., with respect to tumor contrast) [375]. In fact, upon ligand multimerization, also unfavorable pharmacokinetic properties had been observed, such as high non-specific binding or slow wash-out from blood or non-tumor tissues (i.e., background uptake in the liver complicating detection of liver metastasis). To this end, preclinical studies with radiolabeled ligand dimers repeatedly showed increased tumor uptake compared to monomers while maintaining suitable pharmacokinetics [376,377]. Several dimeric αvβ3-targeted constructs, based on the cyclic pentapetide cyclo[RGDxK] (x = f,y) with different PEG-linker modifications, namely ^18^F-Alfatide (^18^F-AlF(NOTA-2P-RGD2)) [378], ^18^F-Alfatide II (^18^F-AlF(NOTA-2P-RGD2)) [379], and ^18^F-FPPRGD2 [380] have since been clinically evaluated. They appeared to have higher sensitivity and specificity for the detection of primary and metastatic tumor lesions than monomers [372]. However, this is ultimately difficult to evaluate, since most studies were performed in small cohorts of patients afflicted with different tumor entities. In addition, data for the direct clinical comparison of the performance of a monomer versus a dimer in the same patient are not available. Clinical evaluation of αvβ3-targeting dimers, often in comparison to ^18^F-FDG in proof-of-concept studies, is ongoing. Recent studies include the imaging of cancer of the esophagus [381], breast [382], and lung [383,384,385,386,387], as well as skeletal [379], brain metastases [388], and radioiodine refractory thyroid cancer [389]. Several of these studies reported higher diagnostic accuracy of αvβ3-targeting dimers than ^18^F-FDG, while in others, comparable performance was reported. The database Clinicaltrials.gov does currently not list large-scale clinical studies that describe PET/CT of ^18^F-Alfatide, ^18^F-Alfatide II, or ^18^F-FPPRGD2.

Tetra- and octameric RGD-based peptides (based on cyclo[RGDyK]) also showed increased tumor uptake and retention. In addition, they displayed a delayed blood clearance, which means that they are more suitable for imaging with longer-lived isotopes, such as Cu-64, to be able to image at the time-point where the highest tumor-to-background ratios are achieved [366,390].

More recently, the development of suitable chelators also allowed the introduction of ^68^Ga-labeled cyclic RGD-based trimers, which resulted in a good compromise between increased tumor accumulation and suitable pharmacokinetics in preclinical studies. Recently, four αvβ3-targeting ^68^Ga-labelled multimeric cyclic RGDfK peptides (one dimer and three trimers [391,392,393,394]) were directly compared [394]. These tracers were based on different chelators (1,4,7,10-Tetraazacyclododecane-1,4,7,10-tetraacetic acid (DOTA), 1,4,7-triazacyclononane phosphinic acid (TRAP), fusarinine-C (FSC), and tris(hydroxypyridinone (THP)), resulting in distinct topological structural arrangements, but they were all neutrally charged at physiologic pH. The authors found a general suitability of all four compounds for the imaging of αvβ3-expressing xenografts, but noticeable differences with regard to total tumor uptake and different tumor-to-normal tissue ratios [394]. In particular, the highest tumor-to-background (tumor:blood, tumor:muscle) ratios were reported for ^68^Ga-TRAP(cyclo[RGDfK])_3_ [391], but the highest tumor uptake was reported for ^68^Ga-FSC(cyclo[RGDfK])_3_ [392]. To date, such systematic comparisons of the pharmacokinetic and pharmacodynamic properties of different multimers of the same peptide sequence remain scarce but provide important insights into options for optimizing integrin-targeted tracers for in vivo imaging. Importantly, the selection of the best tracer will also depend on the needs of the intended clinical application, e.g., if it is more important to achieve the highest possible contrast or the highest possible tumor uptake.

In addition to multimeric probes carrying multiple identical integrin ligands, heterodimeric probes combining integrin ligands with ligands directed to other target structures have been explored (e.g., reviewed in [395]). The rationale behind this strategy is to benefit from the increased affinity and tumor retention of peptide dimers and to tackle the heterogeneity of target expression in distinct tumor types by dual receptor targeting, enabling imaging tumors that express either or both targets [396]. A F-18 labeled construct targeting αvβ3 and NRP-1 via an RGD-ATWLPPR heterodimer showed higher uptake and tumor-to-background ratios in a glioblastoma model than the respective monomers [397]. Another heterodimeric approach, targeting the Gastrin-Releasing Peptide Receptor (GRPR) with a bombesin (BBN) analog ligand and αvβ3 (via cyclo[RGDyK]) [384,398], has recently undergone proof-of-concept clinical evaluation [387,399]. In a small study cohort of 22 breast cancer patients, ^68^Ga-BBN-cyclo[RGDyK] showed significant uptake in primary lesions, axillary lymph nodes, and distant metastases [399]. A direct comparison to monomeric ^68^Ga-BBN PET in 11 of these patients revealed that tumor uptake and lesion detection rates were significantly higher for the heterodimeric ^68^Ga-BBN-cyclo[RGDyK]. Simultaneous evaluation of the Somatostatin-2-Receptor (SSTR2), which was targeted with the peptide TATE, and αvβ3 expression using the heterodimeric PET probe ^68^Ga-NOTA-3P-TATE-cyclo[RGDyK] in 32 patients afflicted with lung cancer or neuroendocrine neoplasms resulted in significantly higher tumor-to-background ratios than the monomeric ^68^Ga-NOTA-TATE, ^68^Ga-NOTA-cyclo[RGDyK], and ^18^F-FDG, respectively [400]. These recent clinical evaluations showed a very good correlation between the PET signal and the expression of αvβ3 and the respective other target determined by immunohistochemistry on tumor cells. Additional heterodimeric/heterobivalent ligands at the preclinical evaluation stage, including PET and SPECT tracers targeting GRPR/αvβ3 or PSMA/αvβ3 [401], were recently reviewed [396,402].

Apart from cancer imaging, a growing body of literature suggests a clinical value of αvβ3-targeted PET imaging to assess post-myocardial infarction angiogenesis, using different agents, including ^99m^Tc-IDA-D-(cyclo[RGDfK])_2_ [403], ^68^Ga-NODAGA-cyclo[RGDyK] [404], ^18^F-Galacto-cyclo[RGDfK] [405,406], and others (see [407] for a review on this topic).

**Nuclear imaging of integrin αvβ6.** Nuclear imaging of αvβ6 has seen important advances during the last years. The frequent and widespread overexpression of αvβ6 in many cancer types [10,121,122,123,124,125,126,127], most prominently in HNSCC and pancreatic cancer, and low to undetectable expression in healthy adult epithelia, renders it an exquisite target to achieve high imaging contrasts. Early progress of αvβ6-targeted imaging was reviewed in [10]. Several radiolabeled tracers have shown promise in preclinical PET and SPECT imaging studies, including linear peptides [355,408] (e.g., ^18^F-A20FMDV2, ligand **13**, Figure 7), cystine knot peptides [409,410,411], and cyclic peptides [358,412,413,414] (e.g., ^68^Ga-Avebehexin (based on the nonapeptide cyclo[FRGDLAFp(*N*Me)K], ligand **14**, Figure 7). Since 2017, four tracers have advanced to studies in humans (Figure 8). The first in-human imaging was reported using the disulfide-bridged peptide ^68^Ga-SFITGv6 (ligand **11**, Figure 7) in two patients in a compassionate use setting [353], after having thoroughly characterized this promising tracer in pre-clinical studies, which can also be labeled with ^177^Lu for targeted radiotherapy. In these first clinical images, specific accumulation of ^68^Ga-SFITGv6 in tumors, but not in inflammatory lesions, had been observed. In a second study, PET/CT scans revealed a good visualization of malignant lesions in non-small cell lung cancer patients, but a benefit over ^18^F-FDG imaging was not observed [415]. The same group recently presented ^68^Ga/^177^Lu-DOTA-SFLAP3 (ligand **12**, Figure 7), which exhibited improved affinity toward αvβ6 and increased tumor-specific accumulation in mice when compared to ^68^Ga-SFITGv6. Moreover, PET/CT imaging of one HNSCC patient was performed, allowing clear visualization of the primary tumor (standardized uptake value (SUV)max: 5.1) and of a lymph node metastasis (SUVmax: 4.1) [354]. Cystine knot PET tracers (“knottin”) were also translated to in-human imaging. Knottins are small (≈4 kDa) peptides characterized by three threaded disulfide bonds arranged in a topological knot, with solvent-exposed bioactive loops extending from the structure [416]. They are considered highly versatile, since their pharmacokinetic properties can be modified by changes in the backbone and the introduction of stabilizers [411]. The authors evaluated RGD-based knottin peptides labeled with ^18^F, ^64^Cu, or ^68^Ga in vitro and in vivo. ^68^Ga-NODAGA-R_0_1-MG and ^18^F-FP-R_0_1-MG-F2, respectively, were imaged in a small number of cervical, lung, and pancreatic cancer patients as well as in patients with idiopathic pulmonary fibrosis. Although the small number of patients did not allow the analysis of the diagnostic efficacy of knottin peptide imaging, PET/CT imaging revealed rapid and sustained accumulation in diseased tissues with relatively low background uptake in healthy organs [416]. The fourth αvβ6-binding PET tracer that was introduced into clinical imaging, ^18^F-αvβ6-BP, is based on the 20-amino acid peptide A20FMDV2 (ligand **13**, Figure 7) and features several large polyethylene glycol (PEG) modifiers in order to mitigate the high lipophilicity-related liver uptake of the neat peptide. ^18^F-αvβ6-BP was preclinically characterized and subsequently utilized for PET/CT imaging in patients with breast, colon, lung, or pancreatic cancer [417]. Significant uptake of ^18^F-αvβ6-BP in both the primary lesion and metastases, correlated to αvβ6 expression and qualified this tracer as another promising candidate for further clinical evaluation of αvβ6 by PET/CT [417]. It will be important to see additional clinical data on the diagnostic accuracy of these αvβ6-targeted tracers. In addition, data on the metabolic stability of these vastly different tracer concepts are scarce. Overall, cyclic peptides are expected to provide a higher metabolic stability than linear peptide or disulfide-bridged peptides [313].

While these results are highly encouraging, there are also ongoing efforts to further optimize tracers, especially with regard to increasing and prolonging tumor uptake and improving pharmacokinetic and washout parameters to reduce background uptake (e.g., in the gut), to ultimately enhance the imaging contrast (Figure 8). One approach relies on the introduction of an albumin binding moiety to the tracer ^18^F-αvβ6-BP, which resulted in an increased blood half-life and tumor uptake, but also in an augmented uptake by other tissues [418]. Another approach builds on the concept of increased affinity and tumor retention of integrin multimers, presenting a ^68^Ga-labeled trimeric conjugate of the αvβ6-specific cyclic pentapeptide SDM17 (ligand **15**, Figure 7) [359]. ^68^Ga-TRAP(SDM17)_3_ featured an affinity that increased from 7.4 to 0.26 nM, an improved selectivity for αvβ6, and a 3-fold higher tumor uptake compared to its monomer ^68^Ga-TRAP(SDM17) [419]. This indicates that multimerization is also a promising strategy in αvβ6 imaging with the potential to improve imaging characteristics over monomeric compounds. Hence, several probe design strategies are currently being preclinically and clinically pursued, with encouraging results for αvβ6-targeted imaging approaches in cancer detection.

**Nuclear imaging of integrin α5β1.** Integrin α5β1 has been suggested for tumor imaging but also as an alternative or complimentary marker to αvβ3 for imaging of the tumor neovasculature, since its expression is upregulated during angiogenesis [116,117,118,420]. As described above, the expression of αvβ3 is not limited to tumor endothelium, but it also occurs in tumor cells and a variety of normal tissues. Following the discovery of several specific and selective ligands for α5β1 [15,57,345], α5β1-targeting probes were recently evaluated in pre-clinical models, including antagonists [18,421], a trimeric peptide [422], and a peptidomimetic [423] tracer for PET imaging. In addition, monomeric and dimeric ^99m^Tc-labeled tracers based on the cyclic heptapeptide (cyclo[phg-isoDGRk], ligand **10**, Figure 7) for SPECT imaging [351,352] were reported. Encouraging early results in tracer development await further follow-up studies with a clear focus on defining the role and significance of α5β1 as a novel candidate for clinical nuclear imaging, especially in terms of imaging tumor angiogenesis but also for other indications, such as rheumatic diseases [424].

**Nuclear imaging of integrin αvβ8.** Integrin αvβ8 is also expressed in tumors, e.g., in HNSCC, non-small cell lung cancer, and prostate cancer [104,108,425]. It is a major activator of TGF-β and associated with pathogenic processes related to its dysregulation, e.g., in angiogenesis, tumor growth, invasion, and metastasis [426]. Very recently, the first PET tracers for selective αvβ8 imaging have been developed and validated in vivo, including monomeric and trimeric variants of cyclo[GLRGDLp(*N*Me)K] (ligand **16**, Figure 7) labeled with Ga-68 via click chemistry [360,426]. In accordance with the advantageous properties of multimeric ligands found for other integrins, the tracer ^68^Ga-Triveoctin, a trimer based on cyclo[GLRGDLp(*N*Me)K], exhibited higher tumor uptake and tumor-to-background ratios in xenografts in mice. Furthermore, the imaging of this tracer in a healthy volunteer indicated a favorable biodistribution for clinical imaging.

The possibility of nuclear imaging of αvβ8 and α5β1 opens new avenues for potential diagnostic and therapeutic applications by targeting these integrin subtypes. However, the successful implementation of such approaches will depend on a better understanding of their role in tumors and their potential as a diagnostic, prognostic, or therapeutic target.

### 5.2. Targeted Radiotherapy (TRT)

The ultimate goal of nuclear medicine, of course, is to visualize *and* treat tumors. One approach to achieve this goal is targeted radiotherapy (TRT). In this approach, the integrin ligand is labeled with a therapeutic isotope (e.g., Lutetium-177, Actinium-225) instead of an imaging isotope, which delivers particulate radiation to cancer cells, inducing cell death. TRT allows the treatment of patients afflicted with metastatic disease, which is not possible with external beam radiotherapy. This concept can also integrate both imaging and therapy, which is then called “theranostics”, where the same ligand is first used for diagnostic imaging and subsequently for TRT. Such a theranostic approach has only been clinically approved so far for neuroendocrine tumor therapy and is in late-stage clinical trials for prostate cancer, using agents targeting the SSTR2 [427] and the Prostate Specific Membrane Antigen (PSMA) [428], respectively. For integrin targeted TRT, tracers labeled with different therapeutic isotopes have been suggested, e.g., ^177^Lu and ^67^Cu. In order to create a therapeutic window, TRT agents should display a suitable biodistribution. In particular, radiation damage in organs such as the kidney, liver, and bone marrow are to be avoided while delivering a sufficiently high radiation dose to the tumor in order to induce a growth delay or regression. Recent reports on TRT agents targeting integrins are sparse. One study reported an αvβ3-targeted ^177^Lu-labeled cyclo[RGDfk] (“^177^Lu-EB-RGD”), which was modified with an albumin-binding domain to increase blood half-life and tumor accumulation [429]. ^177^Lu-EB-RGD treatment led to decreased tumor growth, which could be further amplified by combining TRT with anti-PD-L1 antibody therapy. In non-small cell lung cancer PDX models, treatment with ^177^Lu-EB-RGD led to complete tumor eradication in an αvβ3-high expressing model and a significant growth delay in a αvβ3-low expressing model [430]. In addition to targeting αvβ3, a ^67^Cu-labeled RGD multimer (^67^Cu-cyclam-RAFT-cyclo[RGDfK]_4_ was used to treat U87MG glioblastoma xenografts, leading to a significantly delayed tumor growth compared to the control groups, while no major treatment associated toxicities were observed [431].

To date, there are no reports on TRT approaches for selective integrin ligands to other integrin subtypes. One study describes radiolabeling of the αvβ6-targeting nonapeptide cyclo[FRGDLAFp(*N*Me)K] (ligand **14**, Figure 7) with Lu-177 [414]. Although the Lu-177-labeled tracers exhibited a high target affinity and selectivity, the authors drew attention to the need for optimizing their biodistribution prior to attempting in vivo radiotherapy. In particular, the high kidney uptake and the moderate tumor uptake deemed the TRT agents unsuitable for further evaluation.

It is important to be aware that the promising results from these early TRT studies are not easily transferable to humans, since mice have a much higher tolerance for radiation than humans before they present systemic toxicity. Furthermore, differences in the body size between mice and humans (and the size of the treated tumors) play an important role in TRT, since the therapeutic effect is mediated by DNA damaging particles that have certain path lengths, defining how far they can travel through tissue and induce their damage. Lastly, the ubiquitous expression of αvβ3 will render the translation of TRT agents more difficult, since all αvβ3-expressing organs will be targeted. Nevertheless, clinical translation of integrin-targeted TRT can be expected in the near future. In this regard, αvβ6 currently seems to be the most promising target, due to its overexpression in tumors and the availability of novel ligands with suitable pharmacokinetic profiles.

### 5.3. Non-Nuclear Imaging of Integrins

Non-nuclear imaging approaches may serve as interesting alternatives to whole-body nuclear imaging of integrins in the preclinical and clinical setting. They bear the advantage of omitting patient exposure to radiation but require compromises either with regard to sensitivity, quantification, and/or tissue penetration. Here, we will briefly describe recent advances in integrin-targeted fluorescence imaging, optoacoustic imaging, and Raman spectroscopy. We refer the reader to more extensive reviews on different modalities, including optical imaging [432], magnetic resonance imaging [433,434], and molecular ultrasound [435]. Here, we will present selected, recent highlights from the field of non-nuclear imaging of integrins.

**Fluorescence imaging**. Fluorescence-based approaches to target overexpressed integrins on tumor cells have been explored extensively and have resulted in a large amount of probe constructs and pre-clinical studies to visualize tumors. In the majority of such approaches, cyclic RGD-based peptides have either been directly labeled with a fluorophore or have been used as targeting moiety in fluorescently labeled nanomaterials (see [436,437] for recent reviews), which can also be used to deliver drugs into tumors.

To date, most of these approaches are confined to preclinical imaging. Nevertheless, fluorescence imaging of integrins will most likely also benefit from the novel, more selective integrins ligands for distinct integrin subtypes. For example, integrin targeted probes for fluorescence-guided surgery (FGS) are emerging [438]. In FGS, an overexpressed biomarker is used to visualize tumor cells to achieve safer and more accurate surgical resections. Large-scale clinical trials with fluorescently labeled antibodies, peptides, or small molecules have only begun in recent years [438].

Integrin αvβ3-targeted FGS agents have shown promising results in surgical interventions in pets, such as cats and dogs [439,440]. Since such studies can be conducted in a very similar fashion to clinical phase I studies, they can deliver important data on safety and feasibility for improved lesion visualization of new FGS agents and drive translation.

Integrin αvβ6 could also be an excellent target to guide surgical interventions in tumors, e.g., in HNSCC, where it is highly overexpressed at the tumor border, which is potentially associated with its role in EMT during tumor progression and invasion [441]. Achieving complete resections and reducing the rate of positive tumor margins in HNSCC could significantly decrease disease recurrence and increase patient survival. Fluorescence imaging of αvβ6 using a cyclic peptide [358] that was labeled with Cy5.5 has been suggested [442]. In this cytology-based study, the authors reported a sensitivity of 100% and specificity of 98.3% compared to the cytological finding for the evaluation of bony resection margins of patients with bone-infiltrating HNSCC [442]. A recent histological analysis of HNSCC patient samples, which was aimed at identifying promising targets for fluorescence-guided surgery, also identified αvβ6, together with the EGFR, as promising biomarkers [441]. A fluorescently-labeled knottin-peptide for αvβ6 has also been evaluated for its suitability in fluorescent-guided surgery in a genetically engineered pancreatic cancer model, where the authors found specific uptake and high tumor-to-background ratios [443]. Lastly, recently, the first optical probes based on the A20FMDV2 peptide were reported [444], emphasizing the intense research efforts to further advance nuclear and optical αvβ6-targeted imaging toward clinical translation.

A small number of integrin-targeting probes has already been translated to humans for fluorescence-guided surgery. The RGD-based pentapeptide cyclo[RGDyK] was conjugated to a near infrared dye to generate “cRGD-ZW800-1”. The probe presented suitable in vivo imaging properties in preclinical studies in various tumor models [445] and was translated to first-in-human imaging in colon carcinoma, where the investigators were able to visualize colorectal tumors over normal mucosa and reported no safety concerns of the procedure [446]. Although the authors describe their approach as αvβ6-targeted, we would like to point the reader to studies showing that cyclo[RGDyK] exhibits high affinity to αvβ3 (IC_50_: 3.8 nM), and lower affinity to αvβ6 (IC_50_: 86 nM) [318], suggesting that tumor uptake of cRGD-ZW800-1 cannot be primarily be mediated via αvβ6 integrin.

Integrin αvβ3-targeted peptides have also been used for the coating of ultrasmall (6 nm) core–shell silica NPs for FGS in melanoma [447]. These multimodal, αvβ3-targeted nanomaterials underwent clinical translation within a phase I study using cRGDY-Cy5.5-PEG-C′dots (NCT02106598) for FGS in head and neck melanoma and a phase I trial (NCT01266096) using dual modality NPs, which are labeled with a radioactive isotope (^124^I or ^131^I) and a fluorescent dye (Cy5) in metastatic melanoma patients.

**Optoacoustic imaging**. Optoacoustic imaging is a relatively new optical imaging technique. In optoacoustic imaging with externally administered agents, ligands are conjugated to so-called sonophores, which absorb light, causing molecular vibration and small pressure waves that ultimately generate a thermoelastic expansion. The resulting acoustic waves are detected by sensitive ultrasound transducers and then reconstructed into images using backprojection algorithms [448]. Optoacoustic imaging is able to visualize molecular structures with high resolution in vivo and is often used without contrast agents, to image hemoglobin, lipids, water, melanin, or collagen. Nevertheless, using highly absorbing sonophores could overcome some of the limitations of fluorescence imaging (less scattering, more penetration depths). Recent publications on optoacoustic imaging with integrin-targeting probes indicate growing interest in this field.

In one of the first studies in this field, cyclo[RGDfK] was conjugated to a specific type of sonophore, the so-called quencher, BHQ-1, yielding BHQ-1-cRGD, which was evaluated using a high-resolution optoacoustic imaging technique, termed raster-scan optoacoustic mesoscopy (RSOM) [449]. BHQ-1-cRGD was able to produce αvβ3-specific optoacoustic signals in vitro and in vivo, rendering it a promising candidate for further optoacoustic studies. Other groups have presented similar approaches with promising results. For example, the near infrared dye indocyanice green (ICG) was conjugated to a small cyclic αvβ3-targeted RGD-containing azabicycloalkane peptidomimetic (IC_50_: 53.7 nM) [450]. While the authors were able to detect uptake in U87-MG subcutaneous tumors, but not in A431 tumors, the αvβ3 specificity of the uptake was not determined [451]. Overall, studies on optoacoustic integrin-targeted imaging are in very early stages, and more data are needed to define the potential clinical value of this approach. Currently, these studies are limited to “traditional” integrin-targeting ligands, such as cyclo[RGDfK], but they have not yet expanded to selective peptidic or peptidomimetic ligands for other integrin subtypes.

**Raman spectroscopy**. Surface-Enhanced Resonance Raman Spectroscopy (SERRS) is another innovative modality in which integrin targeting has been explored. cyclo[RGDyK]-SERRS NPs were designed to enable the detection of an early microscopic cancer cell spread from the primary tumor as well as diffuse growth patterns of glioblastoma. This technology has a higher spatial resolution than PET or SPECT and a higher sensitivity than fluorescence imaging. In a pre-clinical study using a genetically engineered murine glioblastoma model, RGD-SERRS NPs showed highly specific, blockable tumor delineation and were able to identify even small clusters of isolated tumors cells [452]. The same group recently reported a novel technology, called Surface Enhanced Spatially Offset Resonance Raman Spectroscopy” (SESO(R)RS), with which they were able to image deep-seated glioblastoma multiforme tumors in vivo in mice through the intact skull [453].

Over time, the molecular imaging of integrins became almost synonymous with angiogenesis imaging and αvβ3-imaging. However, a better understanding of the role of αvβ3 in tumor development and progression and the emergence of imaging probes for other RGD-binding integrins, most prominently αvβ6, are slowly starting to change this perception and approximate a more complex picture of the clinical potential of molecular imaging of integrins. Future emerging probes and clinical studies are necessary to refine and re-define the value of diagnostic imaging of integrins, fully embracing the potential of the variety of integrins and modalities available. Beyond whole-body nuclear imaging for different integrins, targeted radiotherapy and a range of alternative approaches based on fluorescence, optoacoustic, or Raman spectroscopy yield promise for diagnostic and interventional procedures.

## 6. Integrin Targeting in Non-Cancer Diseases

### 6.1. Fatal Role of Integrins in Promoting Sepsis

The progredient state of sepsis is characterized by enhanced vascular permeability and leakiness and thus loss of the barrier function of vessels, allowing the extravasation of pathogenic microorganisms into the blood stream. Consequently, infectious events are disseminated across the whole body, resulting in organ failures especially in the lungs and the urinary tract, and also cause cardiac problems, ultimately provoking a septic shock. Moreover, edema form, which trigger hypoxia as a cause of hampered oxygen supply in tissue areas that are flooded with interstitial fluids [454]. These multiple events represent major challenges in the management of sepsis, together with the development of antibiotic-resistance of bacterial strains. For these reasons, treatment strategies are urgently demanded with a special focus on the improvement of vascular stability. In this context, endothelial cell-expressed integrins come into play, since microbes exploit these adhesion receptors for their early infectious contacts with the endothelium of the inner vessel lining. This is permitted by the presence of an RGD-motif contained within cell surface proteins of many bacteria, viruses, or fungi, which enables them to anchor to integrins and mediates their cellular entry. These events promote vascular dysfunction and ultimately the initiation of apoptotic cell death. This has inspired new therapeutic regimen in order to block the attachment of microbes to the endothelium, thereby preserving cell–cell contacts with undisturbed tight junctions [37,455,456].

One of the most important integrins involved in microbe/endothelial cell attachment and internalization is αvβ3. As described above, this integrin subtype, in concert with growth factors and their receptors, such as VEGFRs, and other molecules, is instrumental for the control of vessel stability. Thus, targeting the αvβ3 docking site for pathogenic microorganisms was conceived for the treatment of sepsis by applying the αvβ3-directed peptide Cilengitide in low doses to ex vivo human vascular endothelial cells under static as well as experimental shear stress conditions. Already the precursor of Cilengitide, the cyclic peptide cyclo[RGDfV] has been shown to have beneficial effects on ischemic acute renal failure in rats pointing to the vascular stabilization effect of the ligand for αvβ3 [457]. Much later, Cilengitide treatment turned out to effectively hinder the attachment of microbes, such as *E. coli* isolated from patients with sepsis [458]. Another study focused on the elucidation of the molecular mechanisms causing endothelial dysfunction after infection with Staphylococcus aureus. Herewith, it was shown that the anchoring of *S. aureus* was mediated by its major cell wall protein Clumping factor A (ClfA), which is capable of binding to endothelial cell-expressed αvβ3 by using blood plasma fibrinogen for bridging to αvβ3. This association was accompanied by Ca^2+^ mobilization, granule exocytosis, and the deposition of the von Willebrand factor (VWF) on endothelial cell surfaces. This provoked finally within 24 h of *S. aureus* attachment a loss of the endothelial barrier function. Concomitant with increased endothelial permeability, also reduced endothelial cell proliferation and increased apoptosis was noticed. Most interestingly, also here, blocking of αvβ3 with Cilengitide reduced endothelial permeability and stabilized vascular endotheial (VE)–cadherin contacts. In vivo*,* the binding of wild-type and ClfA-deficient *S. aureus* Newman to the endothelium was measured in the mesenteric circulation of C57Bl/6 mice and confirmed the in vitro data of the efficacy of low-dose Cilengitide [459].

In a more recent study, the mechanical strength of the trimeric interaction between ClfA, fibrinogen, and αvβ3 was measured on living *S. aureus* by performing single-molecule experiments. These measurements revealed a strong stability of this complex, which was able to withstand applied forces of up to ≈800 pN when compared to the strength of the interaction between integrins and the RGD-motif (≈100 pN). Indeed, the adhesion forces between single bacteria and αvβ3 could be inhibited by either an anti-αvβ3 antibody or by Cilengitide, indicating the involvement of the RGD recognition sequence. Based on these findings, the authors proposed a mechanism for this interaction depending on the elasticity of fibrinogen. If no mechanical stress was applied, the Arg572-Gly573-Asp574-motif contained within the Aα-fibrinogen chains mediates only weak binding to αvβ3. However, under high mechanical stress, other cryptic Aα chain Arg95-Gly96-Asp97 sequences are exposed, which confer strong integrin binding emphasizing a force-dependent binding mechanism between ClfA and αvβ3, which might also be exploited to attack staphylococcal bloodstream infections [460]. Moreover, in earlier studies, it had been shown that host cell adhesion may also involve fibronectin to form a bridge between cell-expressed α5β1 and Fibronectin-Binding Proteins (FnBP) of certain bacteria representing an important prerequisite for their cellular entry. Hereby, one FnBP binds to several fibronectin molecules, which instigates α5β1 clustering followed by the intracellular signaling needed for internalization involving FAK, Src, PI3K, and Akt [461,462,463,464].

In order to explore if Cilengitide would be useful in a post-bacterial attachment situation, clinical strains of *S. aureus* or *E. coli* were perfused over human endothelial cells in a real-time ex vivo model of sepsis. In case Cilengitide was kept present from the beginning of the perfusion, the attachment of *S. aureus* and *E. coli* to endothelial cells was totally abrogated. By adding Cilengitide at a later time point, it also caused a detachment of bound bacteria [465].

In summary, these data underline the efficacy of Cilengitide to serve as a potent antagonist for the binding of bacteria to endothelial cells via e.g., ClfA, thereby impairing the spread of infection culminating in multiorgan failure. Even more, Cilengitide treatment is conceivable as a prophylaxis in high-risk patients [459,465]. The novel pure antagonistic ligands, which we described above, have not yet been tested so far, but they could be even more effective in hindering microbe attachment in sepsis.

### 6.2. Integrins in Fibrosis

Fibrosis is a chronic disease, which finally leads to organ failures. So far, no efficient treatment regimen is available. TGF-β is majorly involved in the pathogenesis of fibrosis. In fibrotic lesions, the spatially restricted generation of bioactive TGF-β from latent stores is required. In addition to a number of proteases capable of activating TGF-β, integrins are well-documented activators of latent TGF-β (see Section 3.1) [466]. Already in earlier studies, it has been shown that αv-integrins trigger TGF-β activation in fibrotic tissues [467,468]. This triggers the exposure of active TGF-β to its receptors so that it can exert its biological functions [469]. In vivo, the mechanosensitive activation of TGF-β in lung fibrosis has been suggested to involve αv-integrins [470].

In fact, the implication of distinct αv-integrins in fibrosis depends on the diseased organ. Integrins αvβ3/αvβ5 are crucial in cardiac fibrosis [471] and αvβ6 is crucial in renal and pulmonary fibrosis [141,472,473,474]. Currently, among the anti-fibrotic αv-integrins, αvβ6 is the best characterized integrin in fibrosis. The αvβ6 targeting antibody BG00011 is currently evaluated in a phase IIa clinical study for idiopathic pulmonary fibrosis (NCT03573505) [110]. An αvβ6-targeting RGD-mimetic, GSK3008348, was recently published as novel, highly selective, and affine αvβ6 ligand with anti-fibrotic therapeutic potential [475,476,477]. GSK300834 reduced downstream pro-fibrotic TGF-β signaling to normal levels by rapid internalization and lysosomal degradation of αvβ6. In a bleomycin-induced murine lung fibrosis model, GSK300834 was used as an inhalable therapeutic and reduced lung collagen deposition and fibrosis progression [475]. The results of a small phase Ib study (eight patients) indicate that nebulized dosing with GSK300834 was safe and led to successful target engagement [478]. Future clinical studies with GSK300834 are eagerly awaited.

Moreover, there is evidence for a function of αvβ8 in small airway fibrosis associated with Chronic Obstructive Pulmonary Disease (COPD) [479], although it is not totally clear yet. Only recently, the first selective inhibitor has become available as experimental tool to study fibrosis (ligand **16**, Figure 7) [360]. Most interestingly, cell types associated with fibrosis, such as lung fibroblasts, have been shown to harbor αvβ1 on their cell surface. Similar to other αv-family members, such as αvβ6 [467] and αvβ8 [132], αvβ1 is also capable of binding to the RGD-motif contained within the amino-terminal fragment LAP of TGF-β1 and TGF-β3, respectively. Interestingly, however, in that study, the deletion of either αvβ3, αvβ5, or αvβ8 in fibroblasts did not affect the symptoms of fibrosis. Proof that αvβ1 could here be a major player for TGF-β activation in fibrosis was obtained by using a specific and highly integrin subtype-selective small molecule inhibitor to αvβ1, which blocked TGF-β activation on fibroblasts from multiple organs. In animal models of pathologic tissue fibrosis, continuous subcutaneous administration of this inhibitor led to a significantly reduced fibrosis, encouraging a new integrin-targeted therapeutic intervention for the blockade of TGF-β activation in fibrosis by use of this inhibitor with expected low undesirable side effects [141].

### 6.3. Integrins and SARS-COV-2

In light of the currently ongoing pandemic of Coronavirus Disease-19 (COVID-19) caused by the Severe Acute Respiratory Syndrome Coronavirus (SARS-CoV-2), we reviewed the literature concerning connections between integrins and SARS-CoV-2. It is well established that RGD-motifs on viral proteins promote their infections by docking to integrins, such as αvβ1, αvβ3, αvβ5, αvβ6, αvβ8, α5β1, α8β1, and αIIbβ3, which are expressed on host cells [40]. If integrins should play a role in the infection process of SARS-CoV-2, integrin ligands could potentially be used in therapeutic approaches to prevent or treat infections.

However, none of the previously known coronaviruses possessed an RGD-binding domain within the spike protein on the virion surface, which is instrumental in the viral cell attachment [480]. For SARS-CoV-2, the Angiotensin Converting Enzyme 2 (ACE2) has been identified as the important host cell surface receptor through which this virus gains entry by interacting with the receptor binding domain (RBD: amino acids 437–508) located on the spike protein. Sequencing of the spike protein now revealed the presence of an RGD-motif within the RBD (403–405: Arg-Gly-Asp), suggesting the possibility of integrin involvement in the cellular docking of SARS-CoV-2.

Hence, several questions are evident, which remain mostly unanswered. The first and most important question relates to the accessibility of the RGD-motif for integrin binding. Should this be confirmed, a number of important follow-up questions immediately arise: (1) Is there a role of distinct integrins in entry of SARS-CoV-2? (2) Given the variety of RGD-binding integrins, what is the selectivity, specificity, and affinity to different integrin subtypes for SARS-CoV-2? (3) Are ACE2 and integrin binding events interconnected or independent mechanisms for viral entry? (4) In case integrins are majorly involved, do integrin-binding ligands (e.g., peptides, peptidomimetics, or antibodies) have the potential to act as therapeutic agents in preventing or treating SARS-CoV-2 infections?

Computational studies suggest that the RGD-motif is not per se accessible for integrin binding, but it only becomes exposed following proteolytic processing and the associated conformational changes within the spike protein after SARS-CoV-2 binding to ACE2 [480,481]. Interestingly, ACE2 also displays a conserved RGD-motif, potentially available for integrin binding and the enhancement of cell adhesion and thus increased infectivity [482]. However, previous studies already showed that this RGD-motif within ACE2 is inaccessible and would require either a conformational shift to be exposed or an RGD-independent binding mechanism. In fact, the spike protein also contains an LDI-binding motif, which is recognized by some of the non-RGD binding integrin subtypes, indicating an even broader range of possible integrin involvement in host-cell entry of SARS-CoV-2 [483]. The option of integrin binding to both ACE2 and/or the SARS-CoV-2 spike protein led to the hypothesis that integrins might act as competitors of ACE2-mediated SARS-CoV-2 binding, thereby affecting viral entry [484]. Alternatively, Dakal proposed that the interaction between integrins and the RGD-motif on the spike protein could be calcium-dependent [485], allowing the inhibition of this interaction by chelation with EDTA leading, to hampered integrin-mediated signaling and viral infection.

Even though several options of a contribution of integrins by an RGD-based host cell entry of SARS-CoV-2 are discussed in the current literature, peer-reviewed experimental data on this topic are still extremely rare.

One study identified binding events of both, SARS-CoV-2 spike protein and ACE2 to α5β1, indicating a role of this binding mechanisms for viral cell entry [486]. Consequently, the α5β1-binding peptide ATN-161 was capable of inhibiting this association, thereby preventing viral infection of cells in vitro. To our knowledge, this was the first study that identified spike protein binding to an integrin subtype and effective blocking of SARS-CoV-2 infection by a synthetic integrin ligand. Therefore, further studies with ATN-161 are urgently awaited. Data from another study, which focused on a myocardial involvement in SARS-CoV-2 infection, also strengthened the notion of a contributing role of several integrins, but particularly of α5β1, in viral docking and cell entry. The authors found an upregulation of the gene encoding α5, and since α5 only dimerizes with β1, α5β1 was suggested as a candidate for SARS-CoV-2-facilitated cell entry [487]. However, this study did not involve experimental testing of this hypothesis. A role of α4 has been implicated in a case study. A patient with multiple sclerosis received the therapeutic α4-targeted antibody Natalizumab, 48 h before the onset of COVID-19 symptoms. Despite suggestive symptoms and an opaque chest x-ray, the patient repeatedly tested negative for SARS-CoV-2. Eventually, only a single RT-PCR was positive (7 days after the onset of symptoms), and the patient recovered within a week. The authors hypothesize that integrin blockade might have inhibited the viral replication rate, causing the repeated negative results according to the PCR analysis for SARS-CoV-2, as well as a benign course of the disease [488]. Further investigations in this direction are also warranted.

Lastly, it is conceivable that molecular imaging of integrins might also be envisaged for SARS-CoV-2 diagnosis and monitoring. One clinical study is currently exploring the clinical value of PET/CT imaging of αvβ3 in COVID-19 patients, using a dimeric cyclic RGD peptide conjugated with DOTA (^68^Ga-DOTA-(RGD)_2_), but the exact peptide sequence is not disclosed (NCT04596943). Another study focuses on PET/CT imaging of αvβ6 by the probe ^18^F-αvβ6-BP (NCT04376593). In a case report, the authors suggest that αvβ6 imaging could be useful to monitor the persistence and progression of lung damage [489]. Nevertheless, more information is needed on the involvement of integrins and integrin ligands in any aspect of SARS-COV-2 infection, disease progression and recovery, before clinically relevant imaging or treatment approaches can be realized.

## 7. Conclusions

Decades of research into the biological functions of integrins have unraveled strikingly complex molecular mechanisms employed in (patho-)physiological settings. Therefore, targeting of integrins as therapeutic approach in e.g., cancer, fibrosis, thrombosis, sepsis, and viral infections has become a large research area with so far moderate success toward clinical translation. Although some integrin-targeted therapeutics have reached the clinic, a decisive breakthrough in cancer treatment has not been achieved yet. In this review, we dissected the two aspects most relevant for the successful development of integrin-targeted therapeutics, namely a better understanding of the involvement of integrin subtypes in pathophysiological processes and the development of peptidic and peptidomimetic ligands that can selectively and specifically target such subtypes.

As the principal requirement to target integrins, it is important to understand the complex receptor-pharmacology and cell surface accessibility, which depends on the cellular integrin internalization, trafficking, and sorting. All of them are fine-tuned mechanisms deciding over the intracellular integrin fate, i.e., recycling versus degradation. Changes of those integrin endocytic routes are thought to majorly contribute to cancer cell properties, such as the regulation of intracellular signaling, cell proliferation, survival, invasion and metastasis, as well as the reorganization of the tumor microenvironment. Thus, in addition to inhibiting integrin signaling with antagonistic compounds, the blocking of integrin trafficking pathways might evolve as an additional strategy to combat cancer. In this regard, endosomes already emerged as important signaling platforms; however, the mode of their regulation and thus the significance and consequences of inhibiting their functions is just currently evolving. In addition to cellular integrin expression, another aspect came into play by the discovery of integrin-carrying (tumor) exosomes, long after the initiation of most integrin anti-cancer trials. These small vesicles help to define the ECM deposited around primary tumors, and they also travel to distant sites to prepare a pre-metastatic niche and thus strongly affect disease progression and organ-specific metastastasis. However, the horizontal transfer of engineered exosomes may also be therapeutically exploited by their integrin-directed travel, uptake, and delivery of drug cargo into recipient cells.

Until recently, most therapeutic approaches involving peptides or peptidomimetics employed ligands that targeted either predominantly αvβ3 or had a rather broad affinity spectrum against several integrin subtypes (e.g., αvβ3, αvβ5, and α5β1). The most prominent example in this regard is Cilengitide, which entered clinical trials as an αvβ3/αvβ5 antagonist for anti-angiogenic therapy in glioblastoma but failed to meet its clinical endpoint. Later, we learned that low-dose Cilengitide can also promote vessel growth, opposite to its intended clinical effect. Subsequently, alternative treatment tactics have emerged, which are currently in preclinical development or clinical testing. Novel, highly selective and affine ligands for single RGD-binding integrin subtypes (especially αvβ3, αvβ5, α5β1, αvβ6, and αvβ8) have been developed by rational design, computational docking studies, structure–activity relationship surveys, and the synthesis of ligand libraries. Promising novel therapeutic approaches are based on improving the delivery of drugs, e.g., chemotherapeutics or checkpoint inhibitors via agonistic actions of integrin ligands. In this regard, the first clinical results of iRGD combination treatment strategies are awaited soon. The full therapeutic potential of other integrin subtype-targeting ligands remains largely to be explored, e.g., with respect to their inhibitory action on integrin signaling and thus disease progression or their successful implementation as selective, target-specific drug-delivery vehicles. In parallel, molecular imaging of integrins has become an important tool to visualize integrin expression in non-invasive whole-body imaging but also to evaluate the potential of integrin ligands for therapeutic approaches, e.g., for radioligand therapy. To date, the molecular imaging of integrins was mostly focused on αvβ3-imaging. A number of αvβ3-targeted tracers have been in clinical studies for several years, but their clinical value has not yet been clearly defined. Initially, αvβ3-tracers were thought to predominantly show tumor vascularization, but it has become apparent that the overexpression of αvβ3 also occurs on tumor cells. More recently, imaging probes for other RGD-binding integrin subtypes have emerged, most prominently for αvβ6, which is overexpressed in distinct tumor entities (e.g., HNSCC and pancreatic cancer). To this end, several tracers targeting αvβ6, which bears excellent prospects as target for cancer diagnosis and radiotherapy, have recently been studied in humans. In conclusion, integrins still maintain their outstanding potential as valid targets and biomarkers for the therapy of cancer and other diseases. Important lessons from past clinical trials and the increased availability of synthetic, subtype specific integrin ligands show great promise for individualized diagnostic and therapeutic approaches in the era of personalized medicine.

## Figures and Tables

**Figure 1 cancers-13-01711-f001:**
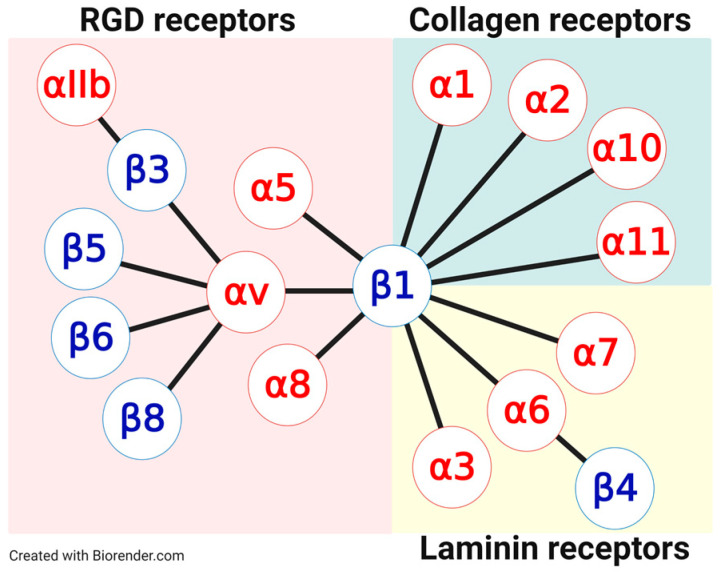
The extracellular matrix (ECM) protein binding integrin family and its classification into subfamilies. The classification is based on the binding specificity of these subfamilies toward ECM proteins. In higher vertebrates, 18 α and 8 β subunits were discovered, forming 24 different integrins through non-covalent interactions. Sixteen of these integrins bind ECM proteins. The leukocyte integrins, cell surface adhesion molecules, are excluded as they are not involved in ECM protein binding but in cell–cell interactions.

**Figure 2 cancers-13-01711-f002:**
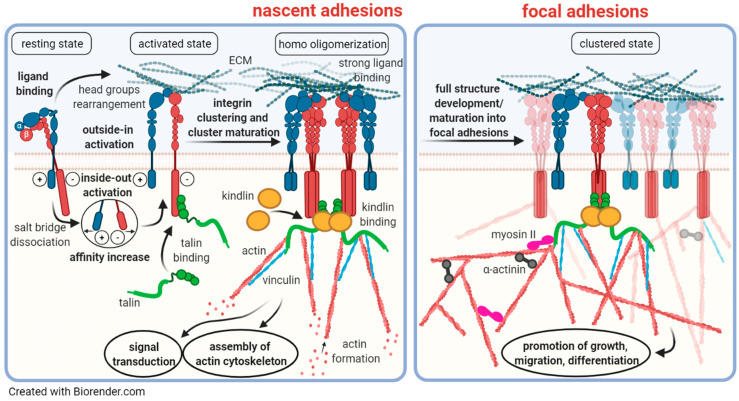
Schematic illustration of integrin activation and the formation of cell adhesion structures. In the resting state, the integrins exhibit a bent conformation. Upon activation, an extended state is formed, the cytosolic salt bridge is disrupted, and the transmembrane helices dissociate. Now, they can homooligomerize to dimers (α-subunit) or trimers (β-subunit), which can further aggregate with other integrins or other proteins, finally forming the highly complex focal adhesions. Accompanied is the intracellular association with proteins such as talin, kindlin, and anchoring to the actin filament. Focal adhesions bind strongly in a multivalent manner to the ECM. RGD-containing ligands can bind to the initial homodimeric state but also to the binding sites of the focal adhesion–ECM complex. Preventing the dissociation of the heterodimers results in pure antagonism, as agonistic activity requires their dissociation of the heterodimeric state.

**Figure 3 cancers-13-01711-f003:**
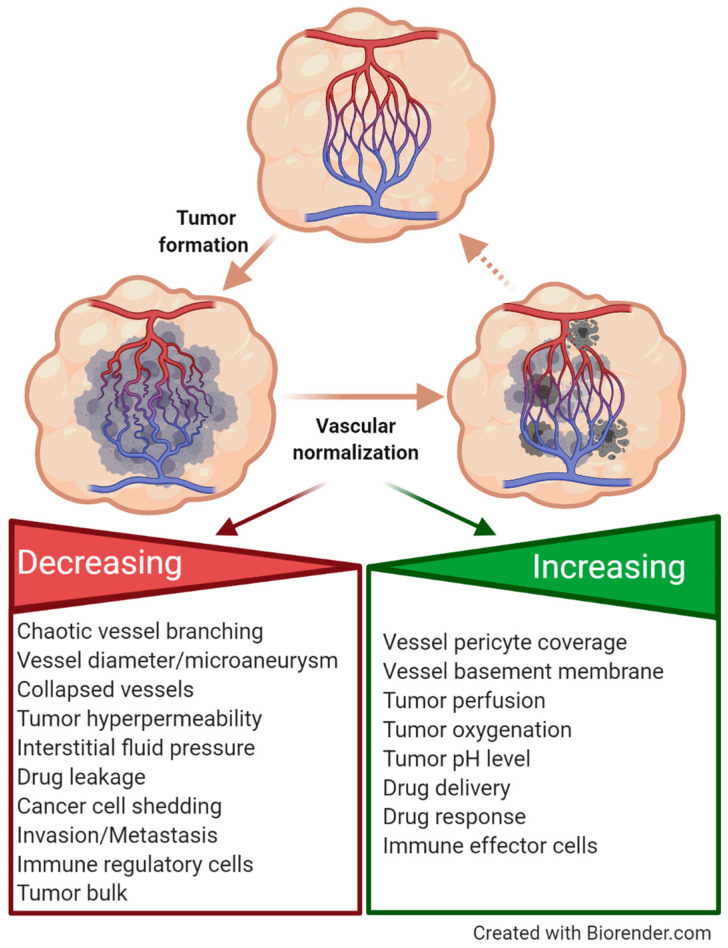
Vascular promotion and vessel normalization provoked by anti-angiogenic therapy. In comparison to the quiescent normal vasculature consisting of mature vessels, the tumor vasculature is structurally and functionally immature. Due to those facts, no regular blood flow is established in tumor vessels, reinforcing a hypoxic environment within the tumors, the extravasation of detached tumor cells into the blood or lymphatic system, and the infiltration by immune regulatory cells. In response to anti-angiogenic therapy, the balance of anti-angiogenesis and pro-angiogenesis is brought back to near normal, achieving vascular normalization. These observations changed the paradigm in cancer therapy from anti-angiogenic strategies to vascular promotion/normalization therapy in order to enhance blood flow and, consequently, increase the delivery and intracellular uptake of chemotherapeutic drugs.

**Figure 4 cancers-13-01711-f004:**
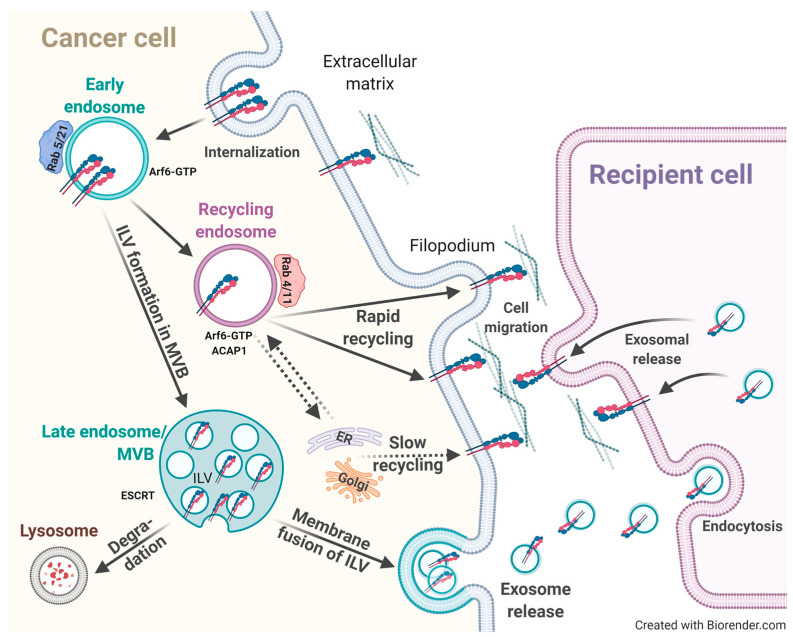
Integrin internalization, sorting, and recycling. Integrin internalization occurs either clathrin-mediated or caveolar involving the dynamin GTPase to form vesicles. Rab GTPase family members and Arf GTPases are intimately involved in integrin trafficking, implicating ACAP1 in rapid integrin recycling from Arf6-positive endosomes. Integrin endocytosis to early endosomes is triggered by Rab5 and Rab21. Integrin recycling proceeds either through a short/rapid-loop by fusion to early endosomes implicating Rab4 signaling, or a long/slow-loop pathway, involving Rab11 signaling, delivering integrins back to the plasma membrane and into filopodia. This facilitates cell adhesion and enhances cell migration. Early endosomes mature to late endosomes/multivesicular bodies (MVBs) via the invagination of the endosomal membrane resulting in intraluminal vesicles (ILVs), which may be prone to degradation in lysosomes. Other receptors are recognized by specific coat proteins and sorted either to the plasma membrane, the trans-Golgi network, or in some cases to the endoplasmic reticulum (ER), thus evading degradation. Constitutive exocytosis of ILVs carrying integrins occurs upon their fusion with the cell membrane. Formed exosomes carrying integrins are taken up by recipient cells where they enrich the cell surface integrin density and thereby increase cell migration. Thus, the proportion between integrins depends on the balance between either their recycling or degradation exerting tremendous consequences for integrin signaling and the biological events arising thereof.

**Figure 5 cancers-13-01711-f005:**
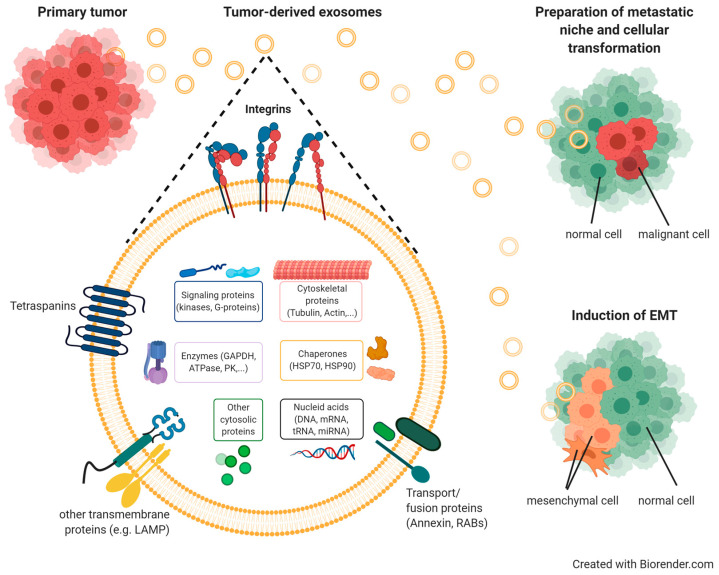
Tumor exosome-mediated intercellular communication and the formation of a pre-metastatic niche. For the final colonization of circulating tumor cells, an appropriate niche at the metastatic site has to be created as well as EMT induced. The formation of such a pre-metastatic niche requires the communication of tumor cells with the local environment of a distant organ by the release of tumor-derived exosomes that deliver their pro-tumorigenic cargo to specifically targeted organs. Depicted is a scheme of a typical exosome that encloses some of the common exosomal components. In addition, integrins are expressed on the exosomal surface, which are capable of anchoring to distinct ECM proteins at the metastatic site.

**Figure 6 cancers-13-01711-f006:**
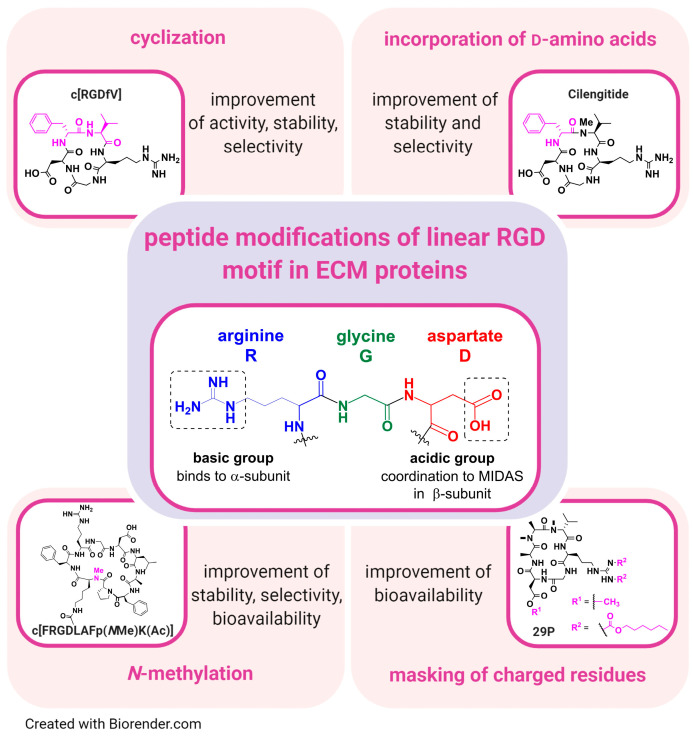
Exemplary peptidic modifications for improving the activity, stability, selectivity, and/or bioavailability of synthetic peptidic integrin ligands. The amino acids RGD coordinate to both integrin subunit: the basic amino acid Arg coordinates to the α-subunit, while Asp coordinates to the Metal Ion-Dependent Adhesion Site (MIDAS) region in the β-subunit. Starting from this linear RGD sequence contained within ECM proteins, various modifications were introduced for improving the integrin targeting properties. The cyclization of peptides and the incorporation of d-amino acids (e.g., cyclo[RGDfV] or Cilengitide) for example improved especially the stability and selectivity of the respective integrin ligands. *N*-methylation (e.g., cyclo[FRGDLAFp(*N*Me)K(Ac)] and the masking of charged residues (carboxylic and guanidinium group in 29P) contributed to a better bioavailability of the named compounds. Furthermore, *N*-methylation led to a higher stability and selectivity of the respective compounds.

**Figure 7 cancers-13-01711-f007:**
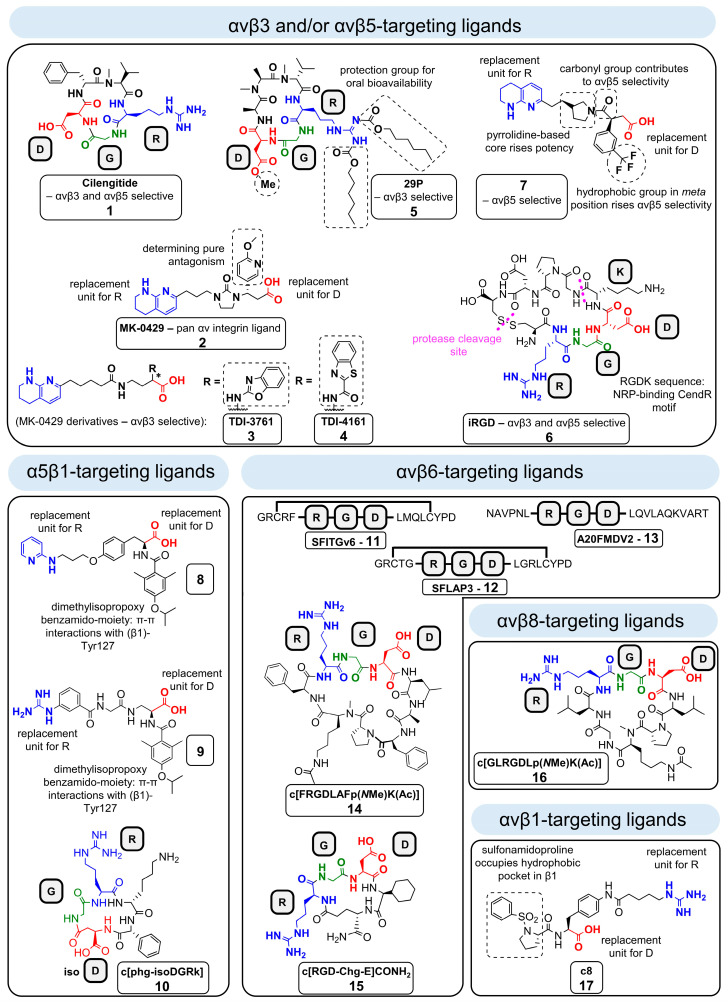
Overview of all described integrin targeting ligands. Within this review, only RGD targeting integrin ligands were described. Hereby, integrins αvβ3, αvβ5, α5β1, αvβ6, αvβ8, and αvβ1, respectively, represent the most important targets for cancer therapy and imaging. phg = d-phenylglycine, Chg = l-cyclohexylglycine, Ac = acetyl, *N*Me = *N*-methylated.

**Figure 8 cancers-13-01711-f008:**
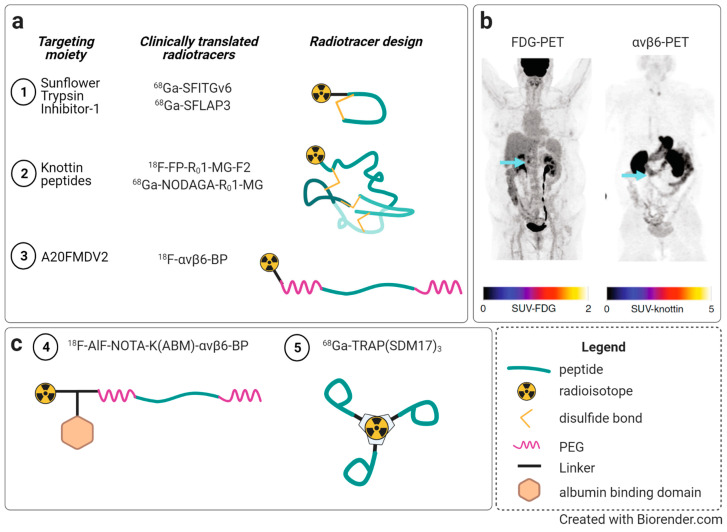
Recent advances in translational αvβ6-targeted Positron Emission Tomography (PET) imaging. (**a**) Clinically translated αvβ6-targeted radiotracers are based on different design approaches, such as disulfide bridged peptides ➀, knottin peptides ➁, and linear peptides ➂. Further descriptions about these tracers are detailed in the text. (**b**) Comparison of [^18^F]FP-R01-MG-F2 PET and [^18^F]-FDG imaging in a patient with pancreatic cancer. The cyan arrow shows tracer accumulation at the head of the tumor (standardized uptake value (SUV)-knottin = 6.3). Several regions of relatively high accumulation were observed, including the kidneys, which are the main clearance route, and the stomach and small intestines, where integrin αvβ6 is expressed. Reprinted from Kimura et al. *Nat Commun*
**10,** 4673 (2019) under Creative Commons CC BY 4.0. (**c**) Design of recent preclinical probes to improve pharmacokinetics and increase tumor uptake. Introduction of an albumin binding domain increased circulation time and tumor uptake ➃ and trimerization of a peptide increased affinity and tumor uptake ➄.

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
