# Peer review of "RGD-Binding Integrins Revisited: How Recently Discovered Functions and Novel Synthetic Ligands (Re-)Shape an Ever-Evolving Field"

_cancers, 2021, doi:10.3390/cancers13071711_

Round 1

Reviewer 1 Report

In the manuscript entitled “RGD-binding Integrins Revisited: How Recently Discovered Functions and Novel Synthetic Ligands (Re-)Shape an Ever-Evolving Field” Ludwig et al. review the progresses about the knowledge of the biological roles of integrins as well as the discovery of new functions in pathophysiological processes in particular in cancer, but also in other diseases, such as sepsis, fibrosis, and viral infections (possibly also SARS-CoV-2). Of particular interest is the paragraph dedicated to integrins internalization, sorting, and recycling and the implication of these processes on ligand binding and therapeutic outcome, as well as the paragraph about exosomal integrins that underline the importance of exosome function in tumor progression and metastasis but also analyze the possibility of their employment in drug delivery applications. Then, under the light of this new findings, the authors analyze/describe the advances in integrin targeted ligands underling the importance of the design of ligands targeted to specific/distinct integrin subtypes and describing their application in biomedical, translational, and molecular imaging approaches focusing on pre-clinical and first-in-human studies as well as on the discovery of novel applications for integrin ligands.

Though, the research about integrins and the design of inhibitors started almost thirty years ago and many reviews about this subject have been published, this field is still alive and the continuous progresses in the elucidation of integrins functions in physiological and pathological processes as well as the efforts in the design of more effective inhibitors/ligands targeting different integrins subtypes deserve to be reviewed.

The review is well structured and the text is well written and fluid. The references are up-to-date and appropriate.

Some minor comments are reported in the following:

  • In figure 1, alpha9 subunit is missing
  • As the authors devote a short paragraph about “Integrin ligands targeting ανβ1”. I think they should also add integrin ανβ1 to the list in section 3.1 "RGD binding integrin receptors".
  • In section 5.3 “Non-nuclear imaging of integrins”, at line 1242, the authors state “For example, integrin targeted probes for fluorescence-guided surgery (FGS) are emerging [442]”. I think that the following studies (not cited by ref 442) concerning integrin targeted fluorescence guided surgery may also be added in the references.

Favril S, Brioschi C, Vanderperren K, et al. Preliminary safety and imaging efficacy of the near-infrared fluorescent contrast agent DA364 during fluorescence-guided surgery in dogs with spontaneous superficial tumors. Oncotarget. 2020;11(24):2310-2326. Published 2020 Jun 16. doi:10.18632/oncotarget.27633.

Wenk CH, Ponce F, Guillermet S, et al. Near-infrared optical guided surgery of highly infiltrative fibrosarcomas in cats using an anti-αvß3 integrin molecular probe. Cancer Lett. 2013;334(2):188-195. doi:10.1016/j.canlet.2012.10.041

  • In section 6.2 ”Integrins in fibrosis” I think that the study by GlaxoSmithKline reporting a small molecule RGD-mimetic, αvβ6 integrin inhibitor, for idiopathic pulmonary fibrosis is worth mentioning

John AE, Graves RH, Pun KT, et al. Translational pharmacology of an inhaled small molecule αvβ6 integrin inhibitor for idiopathic pulmonary fibrosis. Nat Commun. 2020;11(1):4659. Published 2020 Sep 16. doi:10.1038/s41467-020-18397-6;

Maher TM, Simpson JK, Porter JC, et al. A positron emission tomography imaging study to confirm target engagement in the lungs of patients with idiopathic pulmonary fibrosis following a single dose of a novel inhaled αvβ6 integrin inhibitor. Respir Res. 2020;21(1):75. Published 2020 Mar 26. doi:10.1186/s12931-020-01339-7.

Procopiou PA, Anderson NA, Barrett J, et al. Discovery of ( S)-3-(3-(3,5-Dimethyl-1 H-pyrazol-1-yl)phenyl)-4-(( R)-3-(2-(5,6,7,8-tetrahydro-1,8-naphthyridin-2-yl)ethyl)pyrrolidin-1-yl)butanoic Acid, a Nonpeptidic αvβ6 Integrin Inhibitor for the Inhaled Treatment of Idiopathic Pulmonary Fibrosis. J Med Chem. 2018;61(18):8417-8443. doi:10.1021/acs.jmedchem.8b00959.

Author Response

We thank Reviewer 1 for the positive comments and helpful suggestions, which we integrated into the manuscript.

Some minor comments are reported in the following:

  • In figure 1, alpha9 subunit is missing

A: We only display ECM-binding integrin subtypes in figure 1, which applies to 16 of the 24 integrin heterodimers. Alpha-9 is a leukocyte integrin, which does not bind ECM. Therefore, we left it out of this figure. We also state in the figure caption “The leukocyte integrins, cell surface adhesion molecules, are excluded as they are not involved in ECM protein binding but in cell-cell interactions.”

  • As the authors devote a short paragraph about “Integrin ligands targeting ανβ1”. I think they should also add integrin ανβ1 to the list in section 3.1 "RGD binding integrin receptors".

A: We agree with the reviewers suggestion and added a new paragraph (ll. 246-252):

Integrin αvβ1. Integrin αvβ1 has received limited attention as disease-relevant target in the past, possibly a consequence of the absence of specific and selective inhibitory peptides or antibodies. It is highly expressed on activated fibroblasts and directly binds to the latency-associated peptide of TGFβ1 and mediates TGFβ1 activation [1, 2]. Thus, αvβ1 has been suggested as a promising target in fibrosis treatment, but its exact role in TGFβ1 activation, compared to other integrins, remains to be dissected. Integrin αvβ1 has also been described a receptor for viral entry, for example as adenovirus co-receptor.

  • In section 5.3 “Non-nuclear imaging of integrins”, at line 1242, the authors state “For example, integrin targeted probes for fluorescence-guided surgery (FGS) are emerging [442]”. I think that the following studies (not cited by ref 442) concerning integrin targeted fluorescence guided surgery may also be added in the references.

Favril S, Brioschi C, Vanderperren K, et al. Preliminary safety and imaging efficacy of the near-infrared fluorescent contrast agent DA364 during fluorescence-guided surgery in dogs with spontaneous superficial tumors. Oncotarget. 2020;11(24):2310-2326. Published 2020 Jun 16. doi:10.18632/oncotarget.27633.

Wenk CH, Ponce F, Guillermet S, et al. Near-infrared optical guided surgery of highly infiltrative fibrosarcomas in cats using an anti-αvß3 integrin molecular probe. Cancer Lett. 2013;334(2):188-195. doi:10.1016/j.canlet.2012.10.041

A: We agree with the reviewer that these studies should be mentioned. Therefore, we introduced a short additional paragraph (ll. 1253-1258):

Large scale clinical trials with fluorescently labeled antibodies, peptides, or small molecules have only begun in recent years [3].

Integrin αvβ3-targeted FGS agents have shown promising results in surgical interventions in pets, such as cats and dogs [4, 5]. Since such studies can be conducted in a very similar fashion to clinical phase I studies, they can deliver important data on safety and feasibility for improved lesion visualization of new FGS agents and drive translation.

  • In section 6.2 ”Integrins in fibrosis” I think that the study by GlaxoSmithKline reporting a small molecule RGD-mimetic, αvβ6 integrin inhibitor, for idiopathic pulmonary fibrosis is worth mentioning

John AE, Graves RH, Pun KT, et al. Translational pharmacology of an inhaled small molecule αvβ6 integrin inhibitor for idiopathic pulmonary fibrosis. Nat Commun. 2020;11(1):4659. Published 2020 Sep 16. doi:10.1038/s41467-020-18397-6;

Maher TM, Simpson JK, Porter JC, et al. A positron emission tomography imaging study to confirm target engagement in the lungs of patients with idiopathic pulmonary fibrosis following a single dose of a novel inhaled αvβ6 integrin inhibitor. Respir Res. 2020;21(1):75. Published 2020 Mar 26. doi:10.1186/s12931-020-01339-7.

Procopiou PA, Anderson NA, Barrett J, et al. Discovery of ( S)-3-(3-(3,5-Dimethyl-1 H-pyrazol-1-yl)phenyl)-4-(( R)-3-(2-(5,6,7,8-tetrahydro-1,8-naphthyridin-2-yl)ethyl)pyrrolidin-1-yl)butanoic Acid, a Nonpeptidic αvβ6 Integrin Inhibitor for the Inhaled Treatment of Idiopathic Pulmonary Fibrosis. J Med Chem. 2018;61(18):8417-8443. doi:10.1021/acs.jmedchem.8b00959.

A: We agree that the studies above should be mentioned in the fibrosis chapter. Therefore, we introduced the following paragraph (ll. 1428-1437).

An αvβ6-targeting RGD-mimetic, GSK3008348, was recently published as novel, highly selective and affine αvβ6 ligand with anti-fibrotic therapeutic potential [6-8]. GSK300834 reduced downstream pro-fibrotic TGFβ signaling to normal levels by rapid internalization and lysosomal degradation of the αvβ6 integrin. In a bleomycin-induced murine lung fibrosis model, GSK300834 was used as inhalable therapeutic and reduced lung collagen deposition and fibrosis progression [6]. Results of a small phase Ib study (8 patients) indicate that nebulized dosing with GSK300834 was safe and led to successful target engagement [9]. Future clinical studies with GSK300834 are eagerly awaited.

References (different numbering than in manuscript):

  1. Reed, N.I., Jo, H., Chen, C., Tsujino, K., Arnold, T.D., DeGrado, W.F., and Sheppard, D. "The αvβ1 integrin plays a critical in vivo role in tissue fibrosis." Sci Transl Med 7, no. 288 (2015): 288ra79.
  2. Wilkinson, A.L., Barrett, J.W., and Slack, R.J. "Pharmacological characterisation of a tool αvβ1 integrin small molecule RGD-mimetic inhibitor." Eur J Pharmacol 842 (2019): 239-47.
  3. Hernot, S., van Manen, L., Debie, P., Mieog, J.S.D., and Vahrmeijer, A.L. "Latest developments in molecular tracers for fluorescence image-guided cancer surgery." Lancet Oncology 20, no. 7 (2019): E354-E67.
  4. Favril, S., Brioschi, C., Vanderperren, K., Abma, E., Stock, E., Devriendt, N., Polis, I., De Cock, H., Cordaro, A., Miragoli, L., Oliva, P., Valbusa, G., Alleaume, C., Tardy, I., Maiocchi, A., Tedoldi, F., Blasi, F., and de Rooster, H. "Preliminary safety and imaging efficacy of the near-infrared fluorescent contrast agent DA364 during fluorescence-guided surgery in dogs with spontaneous superficial tumors." Oncotarget 11, no. 24 (2020): 2310-26.
  5. Wenk, C.H.F., Ponce, F., Guillermet, S., Tenaud, C., Boturyn, D., Dumy, P., Watrelot-Virieux, D., Carozzo, C., Josserand, V., and Coll, J.L. "Near-infrared optical guided surgery of highly infiltrative fibrosarcomas in cats using an anti-alpha(v)beta(3) integrin molecular probe." Cancer Letters 334, no. 2 (2013): 188-95.
  6. John, A.E., Graves, R.H., Pun, K.T., Vitulli, G., Forty, E.J., Mercer, P.F., Morrell, J.L., Barrett, J.W., Rogers, R.F., Hafeji, M., Bibby, L.I., Gower, E., Morrison, V.S., Man, Y., Roper, J.A., Luckett, J.C., Borthwick, L.A., Barksby, B.S., Burgoyne, R.A., Barnes, R., Le, J., Flint, D.J., Pyne, S., Habgood, A., Organ, L.A., Joseph, C., Edwards-Pritchard, R.C., Maher, T.M., Fisher, A.J., Gudmann, N.S., Leeming, D.J., Chambers, R.C., Lukey, P.T., Marshall, R.P., Macdonald, S.J.F., Jenkins, R.G., and Slack, R.J. "Translational pharmacology of an inhaled small molecule alpha v beta 6 integrin inhibitor for idiopathic pulmonary fibrosis." Nature Communications 11, no. 1 (2020).
  7. Anderson, N.A., Campbell, I.B., Fallon, B.J., Lynn, S.M., Macdonald, S.J.F., Pritchard, J.M., Procopiou, P.A., Sollis, S.L., and Thorp, L.R. "Synthesis and determination of absolute configuration of a non-peptidic alpha(v)beta(6) integrin antagonist for the treatment of idiopathic pulmonary fibrosis." Organic & Biomolecular Chemistry 14, no. 25 (2016): 5992-6009.
  8. Procopiou, P.A., Anderson, N.A., Barrett, J., Barrett, T.N., Crawford, M.H.J., Fallon, B.J., Hancock, A.P., Le, J., Lemma, S., Marshall, R.P., Morrell, J., Pritchard, J.M., Rowedder, J.E., Saklatvala, P., Slack, R.J., Sollis, S.L., Suckling, C.J., Thorp, L.R., Vitulli, G., and Macdonald, S.J.F. "Discovery of (S)-3-(3-(3,5-Dimethyl-1H-pyrazol-1-yl)phenyl)-4-((R)-3(2-(5,6,7,8-tetrahydro-1,8-naphthyridin-2-yl)ethyl)pyrrolidin-1-yl)butanoic Acid, a Nonpeptidic alpha(v)beta(6) Integrin Inhibitor for the Inhaled Treatment of Idiopathic Pulmonary Fibrosis." Journal of Medicinal Chemistry 61, no. 18 (2018): 8417-43.
  9. Maher, T.M., Simpson, J.K., Porter, J.C., Wilson, F.J., Chan, R.B., Eames, R., Cui, Y., Siederer, S., Parry, S., Kenny, J., Slack, R.J., Sahota, J., Paul, L., Saunders, P., Molyneaux, P.L., Lukey, P.T., Rizzo, G., Searle, G.E., Marshall, R.P., Saleem, A., Kang'ombe, A.R., Fairman, D., Fahy, W.A., and Vahdati-Bolouri, M. "A positron emission tomography imaging study to confirm target engagement in the lungs of patients with idiopathic pulmonary fibrosis following a single dose of a novel inhaled alpha v beta 6 integrin inhibitor." Respiratory Research 21, no. 1 (2020).

Reviewer 2 Report

This is a very well written and comprehensive review on RGD-binding Integrins. Starting from generalities, the authors move towards very specific and new aspects concerning therapy. It will be very appreciated by readers because it is original and up to date. The diagrams are particularly well designed and informative.
The only minor comment would be to possibly shorten the title to make it more impactful.

Author Response

We thank the reviewer for the positive evaluation of our manuscript. Since we were not able to find a better, shorter title, we would like to keep our original title.

Reviewer 3 Report

The manuscript is very well written and the reviewer does not have major concern. The only minor issue seems the affiliation of coauthors need to corrected.

Author Response

We thank the reviewer for the positive evaluation of the manuscript. We are not sure what the reviewer refers to with regards to the correction of the affiliations. The affiliations are correct.